# Lipid saturation induces degradation of squalene epoxidase for sterol homeostasis and cell survival

Leng-Jie Huang, Rey-Huei Chen

A fluid membrane containing a mix of unsaturated and saturated lipids is essential for life. However, it is unclear how lipid saturation might affect lipid homeostasis, membrane-associated proteins, and membrane organelles. Here, we generate temperature-sensitive mutants of the sole fatty acid desaturase gene *OLE1* in the budding yeast *Saccharomyces cerevisiae*. Using these mutants, we show that lipid saturation triggers the endoplasmic reticulum–associated degradation (ERAD) of squalene epoxidase Erg1, a rate-limiting enzyme in sterol biosynthesis, via the E3 ligase Doa10-Ubc7 complex. We identify the P469L mutation that abolishes the lipid saturation–induced ERAD of Erg1. Overexpressed WT or stable Erg1 mutants all mislocalize into foci in the *ole1* mutant, whereas the stable Erg1 causes aberrant ER and severely compromises the growth of *ole1*, which are recapitulated by *doa10* deletion. The toxicity of the stable Erg1 and *doa10* deletion is due to the accumulation of lanosterol and misfolded proteins in *ole1*. Our study identifies Erg1 as a novel lipid saturation–regulated ERAD target, manifesting a close link between lipid homeostasis and proteostasis that maintains sterol homeostasis under the lipid saturation condition for cell survival.

## Introduction

Cellular membranes are a complex assembly of lipids and both integrally and peripherally associated membrane proteins (Nicolson, 2014). The lipid composition determines the physiochemical properties of the membrane, such as fluidity, rigidity, permeability, and thickness (Holthuis & Menon, 2014). Membrane fluidity is largely controlled by the ratio of unsaturated to saturated fatty acids and the level of sterols, which determine the packing and the order of the lipids in the membrane (Mouritsen, 2010; Ernst et al, 2016). A fluid membrane permits lateral diffusion of lipids and proteins, which underlies the dynamic events of membranes, such as endocytosis/exocytosis, membrane fusion and fission, and cell division.

Unsaturated fatty acids (UFAs) are converted from saturated fatty acids (SFAs) by fatty acid desaturases, a family of enzymes that introduce cis-double bonds into the acyl chain of acyl-CoA. The critical step is the introduction of the first double bond exclusively between carbons 9 and 10 (C9 and C10, Δ-9 position) by Δ-9 fatty acid desaturase, which converts palmitoyl-CoA and stearoyl-CoA to palmitoleoyl-CoA and oleoyl-CoA, respectively. This reaction requires molecular oxygen and electrons derived from the electron relay systems via cytochrome B5 (Oshino & Omura, 1973). The Δ-9 fatty acid desaturase and cytochrome B5 are expressed as a fusion protein encoded by the *OLE1* gene in the budding yeast, whereas they are produced as separate molecules in mammals (Stukey et al, 1989; Mitchell & Martin, 1995). The mammalian Δ-9 fatty acid desaturase stearoyl-CoA desaturase 1 and yeast Ole1 contain four membrane-spanning regions and are localized at the ER (Bossie & Martin, 1989; Stukey et al, 1989; Bai et al, 2015; Wang et al, 2015).

*OLE1* is essential for life (Henry, 1973; Stukey et al, 1989). *OLE1*-deficient yeast cells fail to transfer mitochondria into the daughter bud, accumulate abnormal membranous structures, and die rapidly (Stewart & Yaffe, 1991). *OLE1* is highly regulated at the levels of transcription and mRNA stability. Increased UFAs repress *OLE1* transcription and promote *OLE1* mRNA degradation (Bossie & Martin, 1989; McDonough et al, 1992; Choi et al, 1996; Gonzalez & Martin, 1996; Vemula et al, 2003). *OLE1* transcription is mediated by two functionally redundant transcription factors Spt23 and Mga2 (Zhang et al, 1999), both of which form homodimers and are kept inactive at the ER through their C-terminal transmembrane domain (Hoppe et al, 2000). Upon ubiquitination by the E3 ubiquitin ligase Rsp5, the protein is cleaved by the proteasome between the cytosolic region and the transmembrane domain to generate a p90 fragment (Hoppe et al, 2000). The processed form is extracted from its unprocessed partner by the ubiquitin chaperone Cdc48-Npl4-Ufd1 complex and migrates to the nucleus to induce the *OLE1* gene (Hoppe et al, 2000; Rape et al, 2001). The saturated and packed lipids alter the rotational orientation of the transmembrane domains of homodimeric Mga2, which promotes its ubiquitination and proteolytic activation (Covino et al, 2016), thereby coupling lipid saturation to *OLE1* expression. Furthermore, the Ole1 protein is short-lived and is degraded by endoplasmic reticulum–associated degradation (ERAD), mediated by the Cdc48-Ufd1-Npl4 chaperone complex (Braun et al, 2002). Together, these regulatory mechanisms ensure an optimal amount of UFAs in cells.

---

Institute of Molecular Biology, Academia Sinica, Taipei, Taiwan

Correspondence: reyhuei@gate.sinica.edu.tw

Besides UFAs, sterol is another determinant of membrane fluidity. As planar molecules, sterols promote the formation of a liquid-ordered state by intercalating between sphingolipids (Ramstedt & Slotte, 2006). Sterols also pack with and order acyl chains, which constrains the motion of acyl chains and decreases fluidity (Ohvo-Rekilä et al, 2002; Czub & Baginski, 2006). Cholesterol and ergosterol are the predominant sterols found in the plasma membrane of animal cells and fungi, respectively. Sterols are essential for life, and their synthesis must be tightly regulated to maintain homeostasis (Jordá & Puig, 2020). Some of the sterol biosynthesis enzymes are subjected to feedback regulation by intermediates in the sterol pathway (Jordá & Puig, 2020). One example is the ER-resident protein 3-hydroxy-3-methylglutaryl coenzyme A reductase (HMG-CoA reductase, HMGCR), the rate-limiting enzyme in the production of mevalonate, the common precursor for sterols and nonsterol isoprenoids. HMGCR is degraded through ERAD when downstream intermediates are increased (Edwards et al, 1983; Hampton & Rine, 1994). The degradation is caused by a reversible conformational change in HMGCR to a more misfolded state (Shearer & Hampton, 2004; Wangeline & Hampton, 2021), leading to its recognition by the E3 ligase of the ERAD (Hampton et al, 1996; Menzies et al, 2018). The second rate-limiting enzyme in the sterol biosynthesis pathway, squalene epoxidase (SQLE, also known as squalene monooxygenase) (Gonzalez et al, 1979), is also controlled by regulated ERAD. Mammalian SQLE is negatively regulated by cholesterol via its N-terminal regulatory domain through the E3 ligase MARCH6 (Gill et al, 2011; Chua et al, 2019), whereas its substrate squalene allosterically stabilizes SQLE by competing for MARCH6 binding to the N-terminal region (Yoshioka et al, 2020). Yeast squalene epoxidase Erg1 is destabilized in response to an elevated level of downstream intermediate lanosterol through the E3 ligase Doa10, the yeast counterpart of MARCH6 (Foresti et al, 2013; Zelcer et al, 2014). The evolutionarily conserved feedback mechanism prevents the accumulation of sterol intermediates that can be toxic to cells (Spanova et al, 2010; Valachovic et al, 2016).

Protein–lipid interaction in the cellular membrane is important for membrane homeostasis. Perturbed lipid metabolism generates lipid bilayer stress, a form of ER stress, which activates the unfolded protein response (UPR) to alleviate the accumulation of misfolded proteins (Volmer & Ron, 2015; Xu & Taubert, 2021). UPR is induced upon the increased saturation of membrane phospholipids either by adding exogenous SFA, by knock downing stearoyl-CoA desaturase 1 in human cells, or by blocking the expression or function of Ole1 in yeast (Wei et al, 2006; Pineau et al, 2009; Ariyama et al, 2010; Surma et al, 2013), with a synergistic effect from ergosterol (Pineau et al, 2009). The ER stress sensors, Ire1 and PERK, can sense the lipid saturation through their ER membrane–spanning domain to activate UPR (Volmer et al, 2013; Halbleib et al, 2017). The loss of very-long-chain fatty acid utilization also increases the membrane saturation and induces UPR, because of a defect in the Ole1 function (Micoogullari et al, 2020). Excess UFA is toxic to cells, unless it is converted to triacylglycerol (TAG) and stored in the lipid droplets (LDs) (Listenberger et al, 2003; Garbarino et al, 2009). Perturbed lipid homeostasis not only causes cell dysfunction and death (Eisenberg & Büttner, 2014) but is also implicated in lipotoxicity-related diseases, such as diabetes and hepatic steatosis (Schaffer, 2016).

Despite the importance of UFA and membrane fluidity, how lipid saturation impacts membrane homeostasis and proteostasis at the cellular level remains largely unknown. Here, we employ temperature-sensitive ole1 mutants to investigate this question and show that Erg1 is targeted for ERAD upon lipid saturation. This protein quality control mechanism ensures sterol homeostasis to maintain membrane integrity and cell survival, showing the importance of the coordinated production of UFAs and sterols for membrane homeostasis.

# Results

## Budding yeast ole1$^{ts}$ mutants are compromised in growth and UFA synthesis

We investigated unsaturated lipid homeostasis in the budding yeast *Saccharomyces cerevisiae*, which has a single fatty acid desaturase gene *OLE1* (Fig 1A). *OLE1* is an essential gene, and its deletion mutant can be kept alive by exogenous oleic acid (OA) (Fig 1B). In order to study the functional role of *OLE1*, we performed error-prone PCR on *OLE1* to screen for temperature-sensitive mutations that suppressed the OA-auxotrophic phenotype at the room temperature (RT, ~25–27°C), but not at 37°C (Fig 1B). Among the collection of temperature-sensitive ole1 mutants, we identified three single-site alleles: ole1-19 (D242G), ole1-20 (S221F), and ole1-40 (F417S) (Fig 1C). All three mutations reside in the cytosolic side, with F417S in the C-terminal cytochrome B5 domain (Fig 1C). We integrated these mutant genes in ole1Δ to replace the knockout selection marker (Fig 1B) and confirmed their auxotrophy for OA at 34°C (Fig 1C). All three mutants grew slowly and almost stopped dividing after 5 h at 34°C (Fig 1D).

Analysis of total fatty acids by gas chromatography–mass spectrometry (GC-MS) showed an increased ratio of the main SFAs (C16:0 and C18:0) to UFAs (C16:1 and C18:1) in all three ole1 mutants compared with the WT at 27°C. The ratio was increased further in the mutants after shifting to 34°C (Fig 1E), indicating that fatty acid desaturase activity was indeed compromised in these mutants. As all three ole1$^{ts}$ mutants appeared to have similar phenotypes, we chose ole1-20 to profile phospholipids by liquid chromatography–mass spectrometry (LC-MS). WT mostly contained at least one unsaturated acyl chain in all lipid classes, whereas the fraction of saturated acyl chains was markedly increased in ole1-20 grown at 34°C (Figs 1F and S1), indicative of membrane lipid saturation.

## Instability of squalene epoxidase Erg1 in *ole1-20*

Increased saturation of membrane lipids in ole1 mutants might impact the structure or function of membrane-associated proteins. To this end, we performed stable isotope labeling using amino acids in cell cultures (SILAC) and found that ole1-20 had a lower level of the squalene epoxidase encoded by *ERG1* than did WT (Fig S2). Immunoblot using anti-Erg1 antiserum confirmed that the Erg1 protein was reduced to ~65% of the WT level in all three ole1 mutants grown at 27°C and reduced further to ~20% at 34°C (Fig 2A).

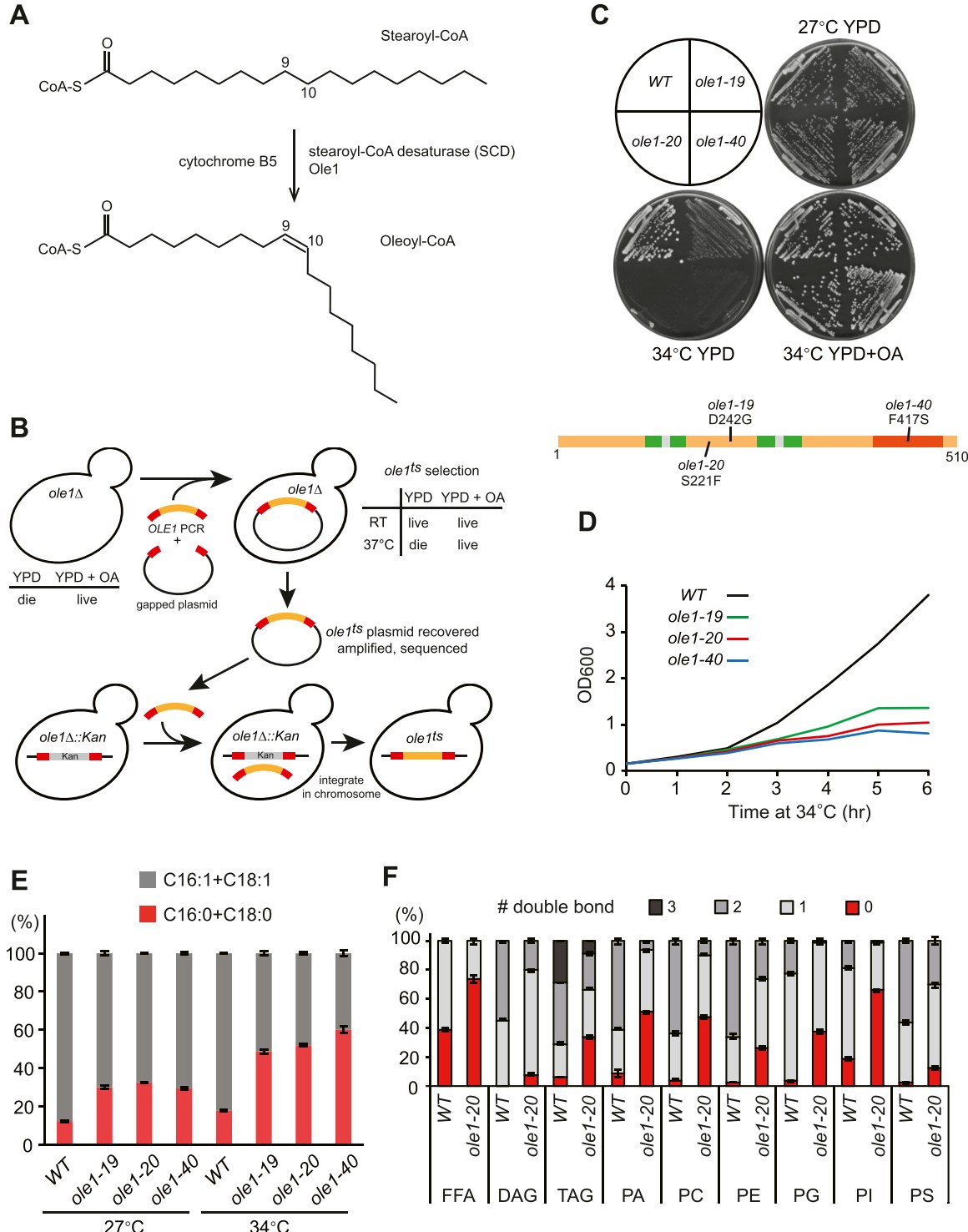

**Figure 1. Temperature-sensitive *ole1* mutants are compromised in cell growth and accumulate saturated lipids.**
**(A)** Conversion of stearoyl-CoA to oleoyl-CoA by stearoyl-CoA desaturase, Ole1 in yeast, and cofactor cytochrome B5. **(B)** Schematic diagram of the screen for temperature-sensitive *ole1* mutants. RT, room temperature (~25–27°C). OA, oleic acid. **(C)** Growth of three temperature-sensitive *ole1* mutants on indicated plates at 27°C or 34°C. The diagram shows the domain structure of Ole1 and the mutation sites of three *ole1* alleles. Green, transmembrane domains; red, cytochrome B5 domain; orange, cytoplasmic region; and gray, ER luminal region. **(D)** Growth rate of WT and *ole1* mutants at 34°C. **(E)** GC-MS analysis of total fatty acids from cells grown at 27°C or shifted to 34°C for 5 h. The percentage of the most abundant SFAs (C16:0 and C18:0) and UFAs (C16:1 and C18:1) is presented as the mean ± SD from three biological repeats. **(F)** LC-MS analysis of free fatty acids and glycerolipids from cells grown at 34°C for 5 h. The percentage of lipids containing indicated numbers of double bonds for each lipid class is shown as the mean ± SD from three biological repeats. DAG, diacylglycerol; TAG, triacylglycerol; PA, phosphatidic acid; PC, phosphatidylcholine; PE, phosphatidylethanolamine; PG, phosphatidylglycerol; PI, phosphatidylinositol; and PS, phosphatidylserine.

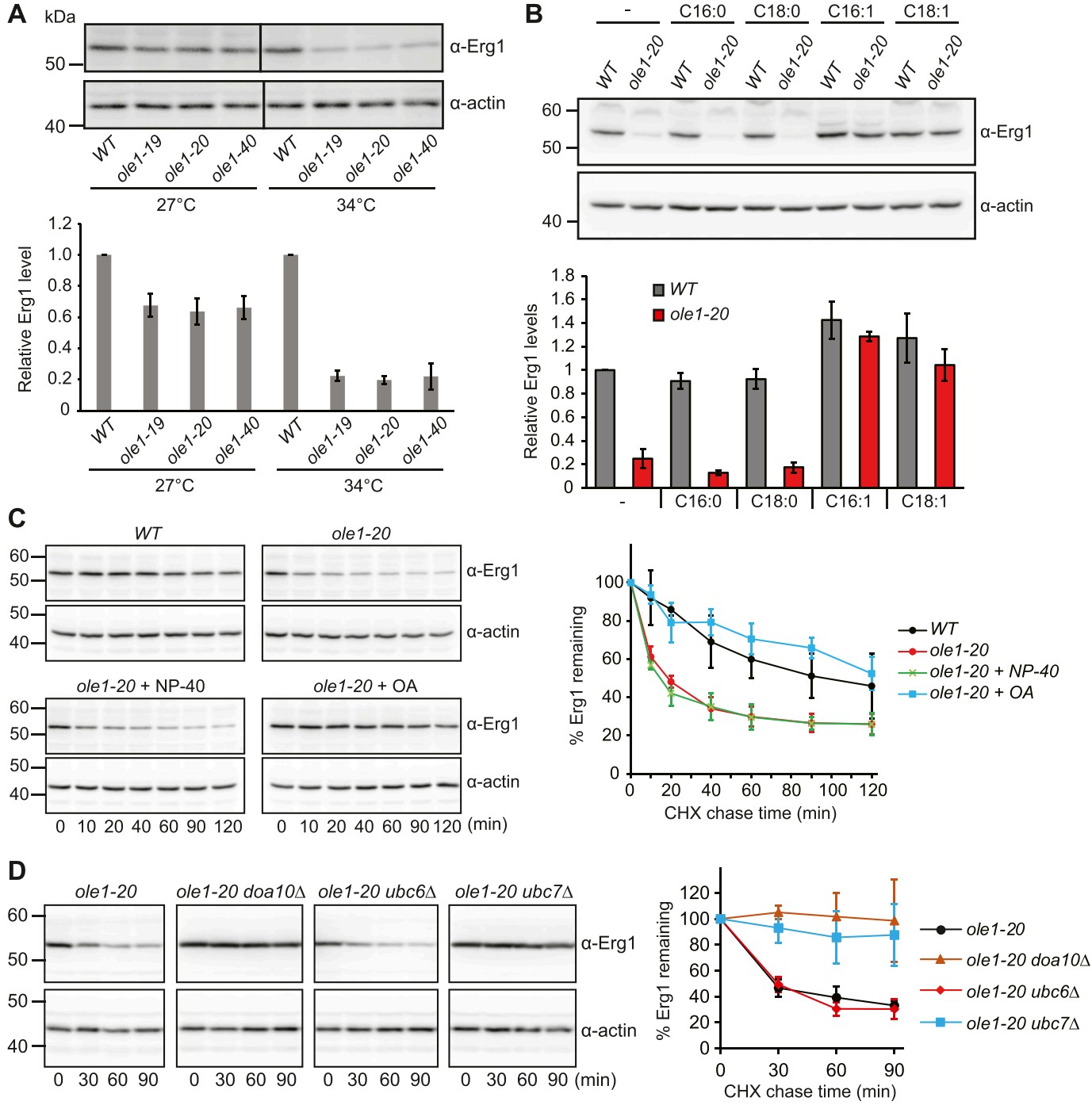

**Figure 2. Erg1 is down-regulated by ERAD upon lipid saturation.**
(A) Immunoblotting of Erg1 in WT and three *ole1* mutants grown at 27°C or shifted to 34°C for 3 h. Actin blot serves as a loading control. Data from three independent experiments are presented as the mean ± SD. (B) Immunoblotting of Erg1 in WT and *ole1-20* grown in media supplemented with the indicated fatty acids at 34°C for 3 h. Data from three independent experiments are presented as the mean ± SD. (C) Turnover of Erg1 upon inhibiting the protein synthesis with cycloheximide in WT, *ole1-20*, and *ole1-20* cells treated with OA or with Tergitol-type NP-40 as a solvent control at 34°C. Data from three independent experiments are shown as the mean ± SD. (D) Turnover of Erg1 in *ole1-20* combined with the deletion of the indicated ERAD genes at 34°C. The graph shows the mean ± SD of three independent experiments.

As Erg1 was similarly reduced in all three mutants, we used *ole1-20* for further studies. Exogenous UFAs, palmitoleic acid (C16:1), and OA (C18:1), but not SFAs, palmitic acid (C16:0), and stearic acid (C18:0), restored Erg1 to nearly WT levels (Fig 2B), indicating that the decreased Erg1 level was caused by lipid saturation. The addition of UFAs also elevated Erg1 levels in WT compared with untreated cells (Fig 2B), suggesting that the increased lipid unsaturation up-regulated the Erg1 level.

Erg1 is known to be a target of ERAD through the E3 ubiquitin ligase Doa10 and E2 enzymes Ubc6 and Ubc7 (Foresti et al, 2013). The cycloheximide chase assay showed that the turnover rate of Erg1 in *ole1-20* was faster than that in WT (Fig 2C), accounting for the decreased level of Erg1. The addition of OA, but not the solvent Tergitol-type NP-40, restored the protein stability (Fig 2C), indicating that Erg1 degradation was triggered by lipid saturation. We next determined whether Erg1 is degraded in *ole1-20* through the known Doa10-Ubc6-Ubc7 complex. The cycloheximide chase assay showed that deleting *DOA10* or *UBC7*, but not *UBC6*, stabilized Erg1 in *ole1-20* (Fig 2D). Together, the results show that the lipid saturation triggers the degradation of Erg1 through Doa10-Ubc7.

### A stabilizing mutation identified in the membrane association domain of Erg1

Erg1 is localized to the ER and LDs (Leber et al, 1998). The available structure of human SQLE, the homolog of Erg1, predicts an atypical membrane association domain at the C-terminus of the protein composed of two α helices (Padyana et al, 2019). The Erg1 structure predicted by AlphaFold2 was similar to SQLE (Fig 3A). We suspected that the C-terminal membrane association domain might be responsible for sensing the lipid environment to control the protein stability. To identify critical residues in this region for Erg1 instability in *ole1-20*, we mutated several residues conserved among Erg1 and its homologs from the fission yeast and human cells (Fig 3B). We ectopically expressed 2myc-tagged Erg1 carrying various mutations and found that the P469L mutant was stabilized in *ole1-20* (Fig 3C). The stabilizing effect was greater than that of the K311R mutation (Fig 3C), which is known to block the ubiquitination of Erg1 (Foresti et al, 2013). Interestingly, Erg1(P469L) showed similar degradation kinetics with Erg1 in WT, indicating that the mutation did not affect the normal turnover of Erg1 (Fig 3C). The results suggest that P469 is critical for lipid saturation–triggered ERAD of Erg1 and that the process likely involves ubiquitination at K311.

We performed molecular dynamics (MD) simulation to gain an insight into how Erg1 associates with the membrane. Simulation with either saturated lipids (PC16:0-18:0, PSPC) or unsaturated lipids (PC16:1-18:1, YOPC) showed that Erg1 is situated on the surface of both types of membranes, with the C-terminal hairpin domain partially embedded in the lipid (Fig 3D). The amino acid P469, residing at the amino-end of the last helix, marked the region furthest in the membrane (Fig 3D). The simulation of Erg1 in lipids mimicking the composition of the membranes in WT and *ole1-20* also revealed similar conformation (Fig S3). The result suggests that Erg1 is not a typical integral membrane protein. Interestingly, a local mass density map from the simulation revealed that P469 coincided with the position of C9 and C10 of the acyl chain in YOPC, but not in PSPC (Fig 3E), showing a spatial correlation between P469 and the double bond of the acyl chain. Furthermore, the differentiation solubilization of isolated microsomes showed that an alkaline solution of 0.1 M $Na_2CO_3$ partially solubilized Erg1, whereas a high salt solution of 0.5 M NaCl released only a small portion of Erg1 (Fig 3F). In contrast, mCherry-tagged Sec61, a multi-pass transmembrane ER protein, remained insoluble in these solutions (Fig 3F).

This result supports that Erg1 is not integrally associated with the membrane.

### Overexpression of stable Erg1 mutants perturbs the ER morphology

We examined the subcellular localization of GFP-Erg1 variants expressed from the *GAL1* promoter and found that WT Erg1, Erg1(P469L), and Erg1(K311R) all localized to the ER marked by the mCherry-tagged ER retention marker HDEL (mCh-HDEL) in WT (Fig 4A). The additional puncta were colocalized with mCherry-tagged Erg6 (Erg6-mCh), a marker for the LD (Fig S4), consistent with the reported distribution of endogenous Erg1 at the ER and the LD (Leber et al, 1998). However, the overexpressed Erg1 and the variants concentrated into large foci in >90% of *ole1-20* cells at 34°C, without typical perinuclear and cortical ER signals (Fig 4A). Interestingly, mCh-HDEL showed a disorganized ER structure in >90% of *ole1-20* cells expressing GFP-Erg1(P469L) and Erg1(K311R), whereas the ER morphology remained normal in ~90% of cells expressing WT Erg1 (Fig 4A). The result suggests that the combined effect of stable Erg1 mutants and saturated membrane lipids causes the aberrant ER morphology.

Lipid saturation might alter Erg1 conformation, leading to its aggregation upon overexpression. We tested this possibility by adding the chemical chaperone glycerol to the growth medium during galactose induction of GFP-Erg1(P469L) at 34°C. Indeed, GFP-Erg1(P469L) now appeared at the perinuclear and cortical ER (Fig 4B). The ER structure was clearly visible in the mid-sections of the cell, although it can be obscured by the foci in the projection image (Fig 4B). Glycerol also restored the ER morphology, as viewed by mCh-HDEL (Fig 4B), suggesting that the accumulation of misfolded proteins might contribute to the disorganized ER. Erg1 normally resides at the ER and the LD, raising the question of whether the foci of the overexpressed Erg1 in *ole1-20* correspond to LDs. Intriguingly, WT Erg1 foci appeared to be distinct from Erg6-mCh foci in *ole1-20* at 34°C, whereas Erg1(P469L) foci were largely colocalized with Erg6-mCh at 34°C and, to a lesser extent, at 27°C (Fig 4C). Pearson's colocalization coefficient also showed a significantly better score for Erg1(P469L) at 34°C than the other two conditions (Fig 4C). It is possible that Erg1(P469L) overexpression in combination with lipid saturation leads to the formation of ER subdomains enriched for Erg1 and Erg6.

### Perturbation of sterol homeostasis underlies the toxicity of Erg1(P469L)

We next asked whether the stable Erg1 mutants might affect cell growth. We found that *ole1-20* cells expressing Erg1(P469L) and Erg1(K311R), but not WT Erg1, from the *GAL1* promoter cannot grow on the galactose-containing plate even at 27°C (Fig 5A), whereas none of these Erg1 variants affected the growth of WT (Fig 5A). The result indicates that the overexpression of stable Erg1 mutants was toxic to *ole1-20*. The toxicity was associated with lipid saturation, because the addition of OA rescued the growth phenotype (Fig 5A).

The toxicity of stable Erg1 mutants in *ole1-20* might result from the perturbed ergosterol biosynthesis pathway. The lipidomic analysis showed that the amount of squalene, the substrate for

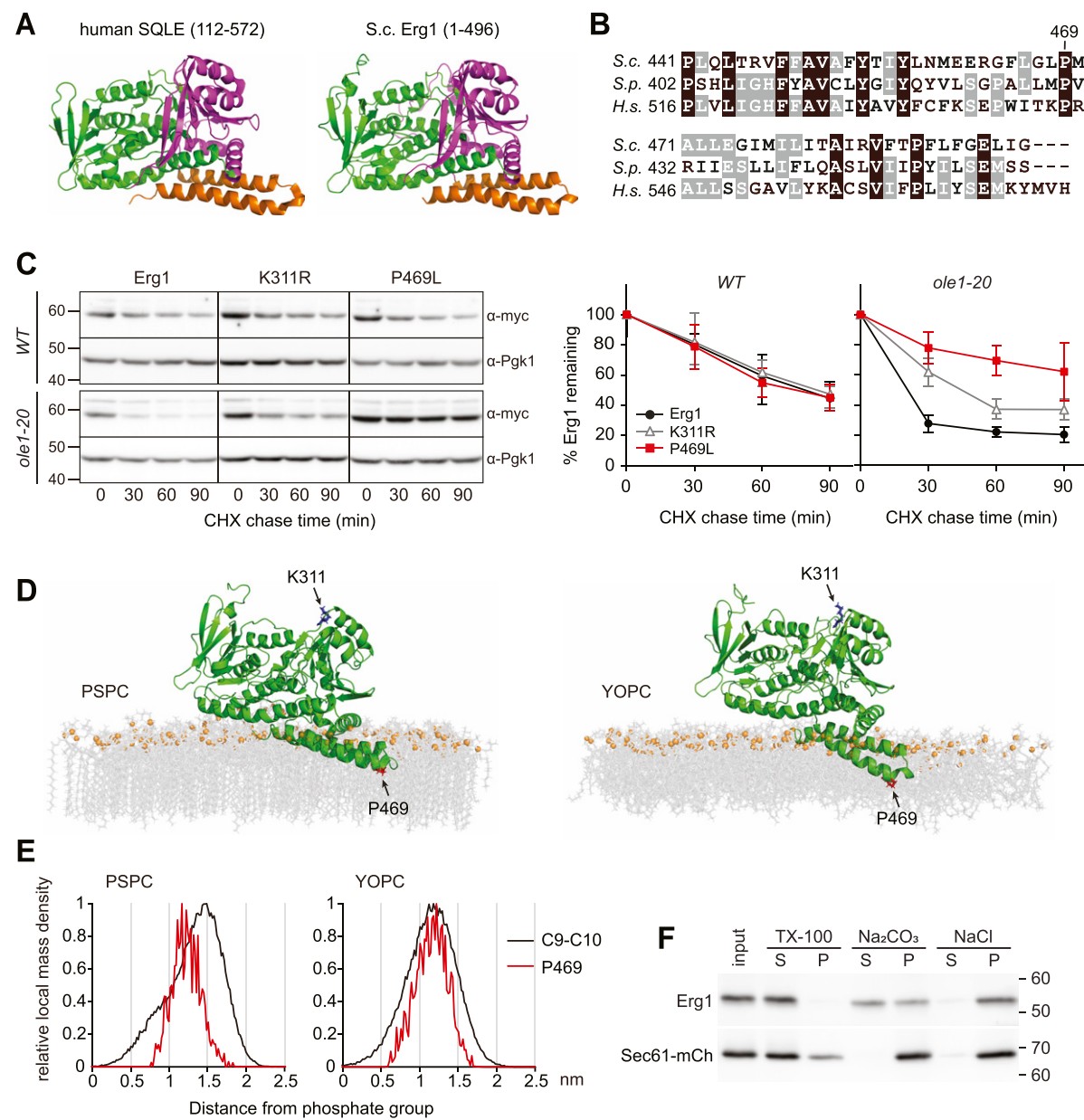

**Figure 3. Residue P469 in the membrane association domain of Erg1 is critical for lipid saturation–triggered degradation.**
**(A)** Human SQLE structure (amino acids 112–572; Padyana et al, 2019) and full-length Erg1 structure predicted by AlphaFold2 are shown in ribbon representation with the cofactor-binding domain in green, substrate-binding domain in magenta, and the C-terminal membrane-associated hairpin domain in orange. **(B)** Sequence alignment of the membrane association domain from the budding yeast (*S.c.*), fission yeast (*S.p.*), and human (*H.s.*) homologs of Erg1. Residues that are conserved among all three species are shaded in black, and those conserved in two species are shaded in gray. **(C)** Turnover of 2myc-tagged WT Erg1, K311R, and P469L mutants in cells after cycloheximide addition at 34°C. Erg1 was detected with anti-myc antibody. Pgk1 blot serves as a loading control. Data from three independent experiments are shown as the mean ± SD in the graph. **(D)** Molecular dynamics simulation of Erg1 in saturated lipids PSPC and unsaturated lipids YOPC. Green, Erg1; gray, acyl chain and head group; and orange, phosphate group. P469 and K311 are highlighted in red and blue, respectively. **(E)** Local mass density of carbons 9 and 10 (C9 and C10) in the acyl chain and P469 in Erg1 determined from the molecular dynamics simulation. The x-axis represents the distance of the target molecules from the phosphate atoms on the lipid layer near Erg1. **(F)** Differentiation solubilization of microsomes isolated from WT using indicated solutions. S, soluble fraction; P, insoluble pellet. Erg1 and Sec61-mCh were detected with anti-Erg1 and anti-mCherry antisera, respectively.

Erg1, was decreased by >30-fold in WT upon GFP-Erg1(P469L) induction from the *GAL1* promoter (Fig 5B), indicating that Erg1(P469L) was enzymatically active. Squalene was increased by 26-fold in *ole1-20* compared with that in WT (Fig 5B), consistent with the loss of Erg1 by degradation. However, Erg1(P469L) overexpression decreased squalene only by <50% in *ole1-20* (Fig 5B), suggesting that the enzyme might not be fully functional in *ole1-20*. Glycerol addition decreased squalene by ~60% in *ole1-20* expressing Erg1 (P469L) compared with that without glycerol addition (Fig 5B), likely because of the restoration of Erg1(P469L) conformation and

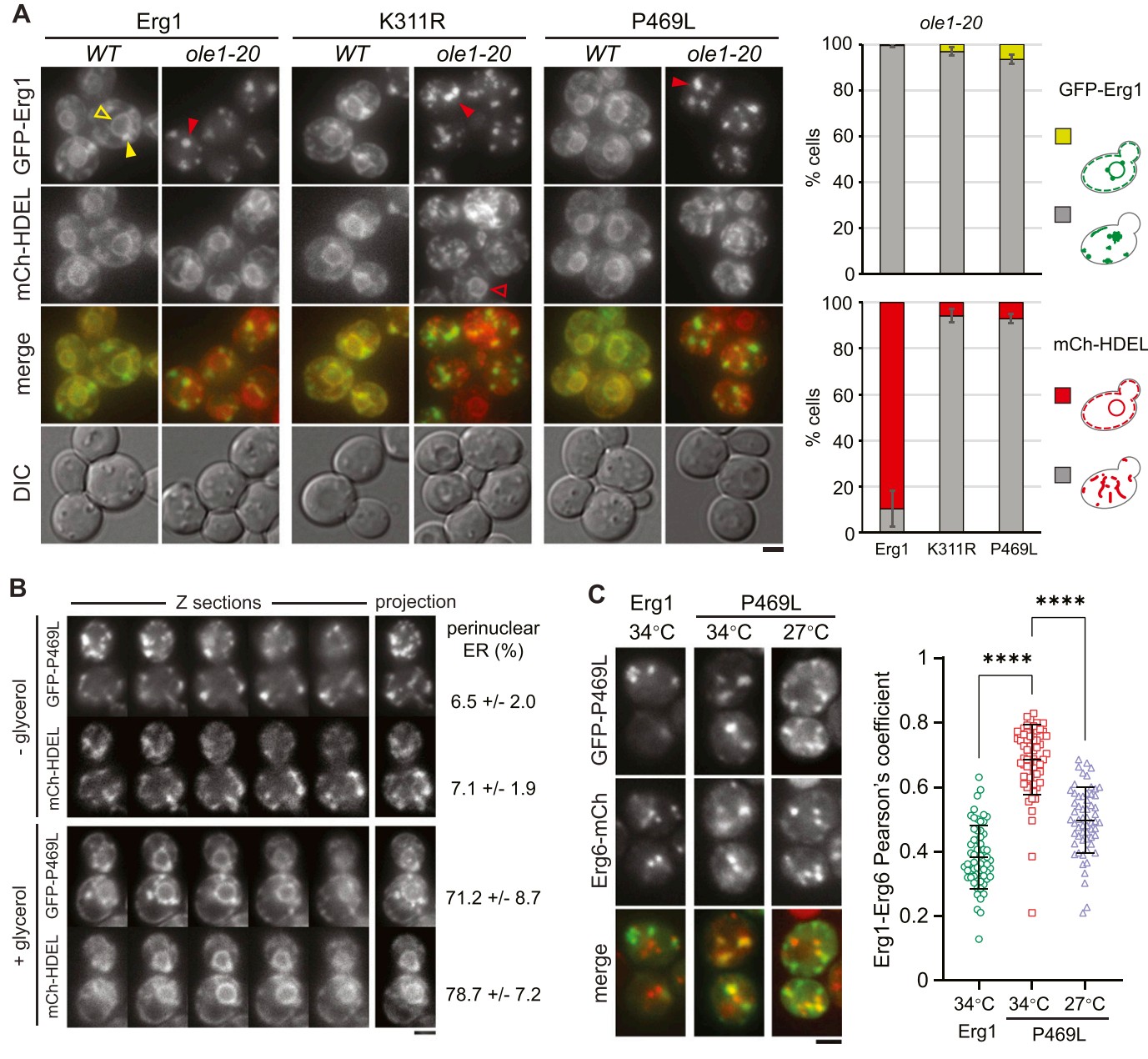

**Figure 4. Overexpressed stable Erg1 mutants form foci and disrupt the ER morphology.**
**(A)** GFP-tagged Erg1, Erg1(K311R), and Erg1(P469L) were expressed from the *GAL1* promoter at 34°C for 3 h in WT and *ole1-20* cells co-expressing mCh-HDEL. Projections of Z-stack fluorescence images are shown. Scale bar, 2 μm. The phenotypes of aberrant protein distribution in *ole1-20* were scored and presented as the mean ± SD from three experiments (n > 200 cells for each sample). Open yellow arrowhead, typical perinuclear signal of GFP-Erg1; closed yellow arrowhead, GFP-Erg1 puncta at the LD; closed red arrowhead, representatives of large GFP-Erg1 foci; and open red arrowhead, normal perinuclear mCh-HDEL in an *ole1-20* cell without K311R expression. **(B)** GFP-Erg1(P469L) expression was induced with galactose in *ole1-20* in the absence or presence of 10% glycerol at 34°C for 3 h. Z-sections and projection images are shown. Scale bar, 2 μm. Cells were scored for the perinuclear ER signal of GFP-Erg1(P469L) and mCh-HDEL and presented as the mean ± SD from three experiments (n > 200 cells for each sample). **(C)** GFP-Erg1 or GFP-Erg1(P469L) was induced in *ole1-20* expressing Erg6-mCh at the indicated temperatures. Projection images are shown. Scale bar, 2 μm. Pearson's colocalization coefficient was determined and is shown as the median ± SD (n = 60 cells for each). ****P < 0.0001 (t test).

subcellular distribution. Surprisingly, the level of ergosterol was not elevated upon Erg1(P469L) overexpression in either WT or *ole1-20* (Fig 5B), indicating that cells were able to maintain an optimal and critical level of ergosterol by regulating other steps in the sterol biosynthesis pathway. Strikingly, Erg1(P469L) overexpression caused the sevenfold accumulation of lanosterol, the first sterol

molecule in the pathway, in *ole1-20*, but only 1.5-fold in WT (Fig 5B), whereas further downstream intermediates were generally decreased (Fig S5). The results suggest that *ole1-20* failed to metabolize lanosterol upon an increased flux of upstream precursors, perhaps because of the down-regulation of lanosterol 14-α-demethylase encoded by *ERG11*. Unlike Erg1(P469L), the WT Erg1

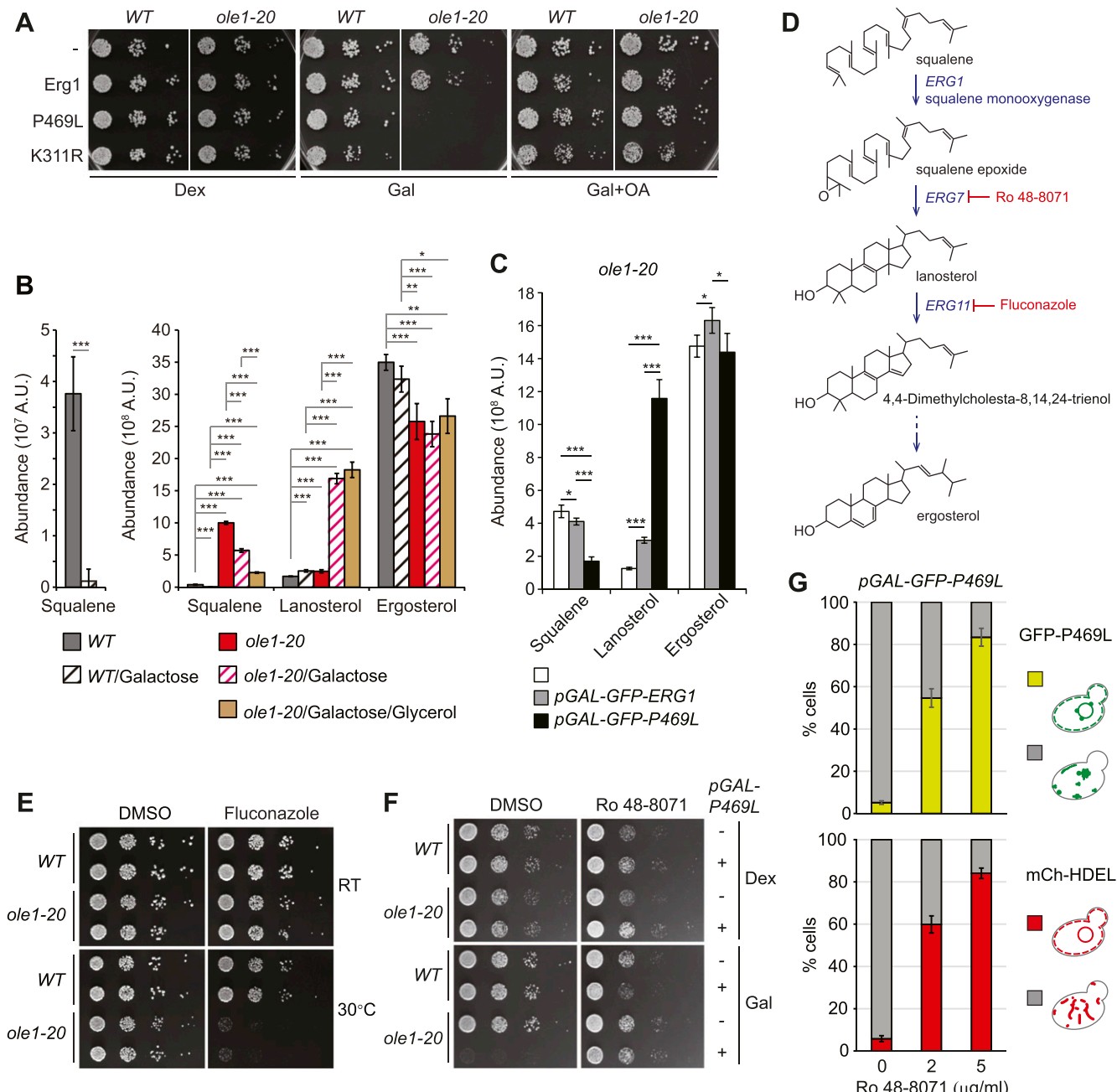

**Figure 5. Erg1(P469L) overexpression is toxic to *ole1-20* because of lanosterol accumulation.**
**(A)** Cells expressing GFP-tagged Erg1, Erg1(K311R), and Erg1(P469L) from the *GAL1* promoter were spotted in 10-fold serial dilutions on YEP plates containing dextrose (Dex), galactose (Gal), or galactose with OA (Gal + OA) and incubated at RT. **(B)** Abundance of squalene and two major sterol species, lanosterol and ergosterol, in cells without or with GFP-Erg1(P469L) induction by galactose at 34°C for 3 h. Glycerol was added together with galactose as indicated. Data from four independent experiments are shown as the mean ± SD. *$P < 0.05$, **$P < 0.01$, and ***$P < 0.001$ ($t$ test). **(C)** Abundance of three sterol species in *ole1-20* without or with galactose-induced WT Erg1 or Erg1(P469L) at 34°C for 3 h. Data from three independent experiments are shown as the mean ± SD. *$P < 0.05$ and ***$P < 0.001$ ($t$ test). **(D)** Steps in the sterol biosynthesis pathway relevant to the study. The dash arrow indicates multiple steps. The inhibitory molecules are indicated in red. **(E)** Serial dilutions of cells were spotted on YPD plates containing 10 μg/ml fluconazole or the solvent control DMSO and incubated at RT or 30°C. Duplicates of each cell type are shown. **(F)** Serial dilutions of cells without or with galactose-inducible GFP-Erg1(P469L) were spotted on Dex or Gal plates containing DMSO or 200 ng/ml of Ro 48-8071 and incubated at RT. **(G)** *ole1-20* cells were pretreated for 20 min with DMSO or with 2 or 5 μg/ml of Ro 48-8071, followed by the addition of galactose to induce GFP-Erg1(P469L) expression at 34°C for 3 h. Cells were scored for the perinuclear ER signal of GFP-Erg1(P469L) and mCh-HDEL and presented as the mean ± SD from three experiments (n > 200 cells for each sample).

elevated lanosterol level by only ~twofold (Fig 5C), likely because of the instability of Erg1 in *ole1-20*. To determine whether the large increase in lanosterol might contribute to the toxicity of Erg1(P469L)

overexpression, we used fluconazole to inhibit Erg11(Fig 5D). We found that *ole1-20*, but not WT, was sensitive to a low concentration of fluconazole at the semi-permissive temperature of 30°C (Fig 5E),

supporting that the accumulation of lanosterol was toxic to *ole1-20*. In addition, inhibiting the lanosterol synthase Erg7 with Ro 48-8071 suppressed the lethality of *ole1-20* from Erg1(P469L) overexpression (Fig 5F), showing that lowering the lanosterol level rescued cell growth. Furthermore, Ro 48-8071 restored the ER distribution of GFP-Erg1(P469L) and the ER morphology in a dose-dependent manner (Fig 5G), suggesting that lanosterol accumulation contributes to Erg1 clustering and aberrant ER phenotypes. Together, these results indicate that the overexpression of stable Erg1 mutants was toxic to *ole1-20*, because of the perturbed lanosterol metabolism.

### *DOA10* deletion recapitulates the phenotypes of Erg1(P469L) overexpression

We next determined whether stabilizing Erg1 in ERAD mutants also produced the same phenotypes as those of Erg1(P469L) over-expression. GFP-Erg1 expressed from the native promoter showed typical ER distribution in *doa10Δ* cells, whereas the protein was enriched in foci in ~70% of *ole1-20 doa10Δ* grown at 34°C for 3 h (Fig 6A and B). The ER marked by mCh-HDEL or Sec61-mCh showed the aberrant morphology without the clear perinuclear ER (Fig 6A and B). Glycerol addition restored the perinuclear and cortical ER morphology and the distribution of GFP-Erg1 at the ER (Fig 6A and B). The phenotype indeed recapitulated that associated with the overexpression of stable Erg1 mutants in *ole1-20* (Fig 4). Furthermore, the nuclear pore complex (NPC) component Nup2 tagged with GFP formed several foci or dispersed in the cytoplasmic ER in *ole1-20 doa10Δ*, instead of evenly distributing around the nucleus (Fig 6A and B). Glycerol addition also restored the nuclear distribution of Nup2 (Fig 6A and B). Thus, blocking the Doa10-mediated protein degradation in *ole1-20* impacts the ER morphology and the distribution of at least a subset of membrane-associated proteins, likely because of the accumulation of misfolded proteins. The *ubc7Δ ole1-20* double mutant also exhibited the Erg1 clustering, aberrant ER morphology, and mislocalization of Nup2 after growing at 34°C for 5 h, whereas *ubc7Δ* lacked these phenotypes (Fig S6). Together, the results suggest that Doa10-Ubc7 is involved in clearing misfolded proteins upon lipid saturation to preserve membrane integrity.

Tetrad analysis of heterozygous *ole1-20* and deletion mutants of ERAD components showed that both *ole1-20 ubc7Δ* and *ole1-20 doa10Δ* double mutants had severe growth defect (Fig 6C), whereas *ubc6Δ* and *hrd1Δ*, E3 ubiquitin ligase of the other ERAD branch, did not show genetic interaction with *ole1-20* (Fig 6C), showing that the degradation of misfolded or defective proteins through Doa10-Ubc7 is important for *ole1-20* survival. We next asked whether the growth defect of *ole1-20 doa10Δ* and *ole1-20 ubc7Δ* double mutants was also attributed to lanosterol accumulation as for Erg1(P469L) overexpression. The double mutants were unable to grow at 32°C, a semi-permissive temperature for *ole1-20* (Fig 7D). The addition of Ro 48-8071 rescued the growth of the double mutants at both RT and 32°C (Fig 7D), indicating that lanosterol accumulation was toxic to the double mutants. Furthermore, Ro 48-8071 restored the distribution of GFP-Erg1, mCh-HDEL, and Nup2 in *ole1-20 doa10Δ* in a dose-dependent manner (Fig 6E). Together, the results show that

ERAD deficiency recapitulates the phenotypes from the over-expression of stable Erg1 mutants. The phenotypes were caused by a combined effect of lanosterol accumulation and lipid saturation, which were suppressed by treatment with either Ro 48-8071 or OA (Figs 6D and E and S7).

The transmission electron microscopy revealed that ~50% *ole1-20 doa10Δ* cells contained cytoplasmic ER extensions, which formed stacks or whorls near or around the nucleus (Fig 7B–D). Unlike the oval or round-shaped nucleus in WT (Fig 7A), the nuclear envelope in some *ole1-20 doa10Δ* cells showed low curvature in some areas and high curvature in others, giving irregular shape (Fig 7B–D). In addition, we observed uneven intermembrane space, closely juxtaposed outer and inner nuclear membranes, and membrane whorls in the intermembrane space in the double mutant (Fig 7B). Strikingly, unlike the visible NPCs in the nuclear envelope of WT, the number of distinguishable NPCs decreased in the double mutant (Fig 7B–D), although a stretch of electron-dense materials may appear in some region of the nuclear membrane (Fig 7Bc). In line with the fluorescence microscopy results, the addition of glycerol in the culture medium suppressed the formation of cytoplasmic ER extensions and restored the nuclear shape and NPCs in *ole1-20 doa10Δ* (Fig 7C and D). Together, the results suggest that ERAD deficiency causes the accumulation of misfolded proteins and lanosterol in *ole1-20* (Fig 8). The combination of ER stress and perturbed membrane lipids results in the aberrant ER morphology and membrane protein distribution (Fig 8).

## Discussion

The composition of lipids and proteins is important for the structure and function of membrane organelles, and for membrane microdomains of specific functions. A shared nature of all membranes is fluidity, which is determined by the unsaturated lipid and sterol contents in the membrane. We have used temperature-sensitive *ole1* mutants to investigate how lipid saturation impacts the lipid homeostasis, proteostasis, and membrane morphology. We demonstrate that lipid saturation triggers the degradation of Erg1 through Doa10-mediated ERAD. The degradation prevents lanosterol, a sterol intermediate downstream of Erg1, from accumulating in cells to avoid toxicity. The research reveals the importance of a close link between UFA and sterol production in the maintenance of membrane homeostasis and proteostasis. The stability of squalene epoxidase presents a key control mechanism for maintaining an optimal sterol/phospholipid ratio.

Lipidomic analysis reveals a global increase in saturation in total fatty acids, free fatty acids, and glycerolipids in *ole1* mutants compared with WT, even when cells were grown at a permissive temperature. The lipid saturation in *ole1* mutants increases further after a shift to 34°C, to a different extent for various lipid classes. Notably, saturation is markedly increased for PC, but not for phosphatidylethanolamine, phosphatidylglycerol, phosphatidylinositol, and phosphatidylserine, suggesting that PC might have a higher turnover rate than other phospholipids. As PC is the most abundant phospholipid and the predominant lipid of cellular membranes (Vance, 2015), a high turnover rate of PC may allow for

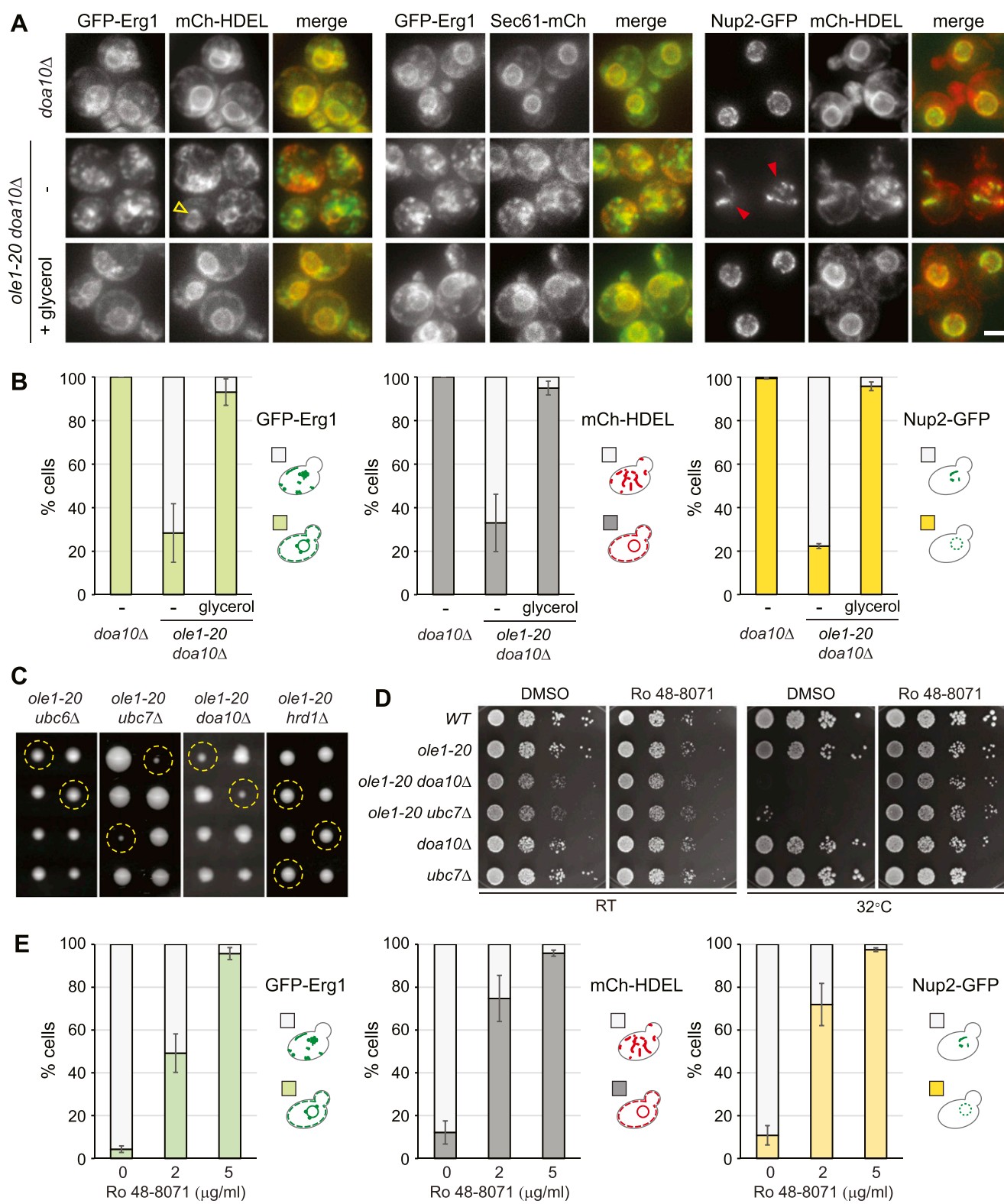

**Figure 6. Doa10 is required for ER integrity in *ole1-20*.**

**(A, B)** *doa10Δ* and *ole1-20 doa10Δ* cells expressing indicated GFP- or mCh-tagged proteins were grown at 34°C for 3 h, in the absence or presence of glycerol. Projection images are shown. Scale bar, 2 μm. **(B)** Cells were scored for the phenotypes described in the diagram and are shown as the mean ± SD, from three experiments for GFP-Erg1 and mCh-HDEL and two experiments for Nup2-GFP (n > 200 cells for each sample). Open yellow arrowhead indicates the perinuclear mCh-HDEL signal in an *ole1-20 doa10Δ* cell, whereas other cells in the field show a disorganized pattern without the discernible perinuclear signal. Closed red arrowheads indicate clustered Nup2 in the cytoplasmic ER. **(C)** Two sets of tetrads each from the indicated heterozygous double mutants are shown. Yellow dash circles indicate the spores carrying double

membrane remodeling to maintain an optimal membrane fluidity during homeoviscous adaptation in response to changes in the environmental temperature (Hazel, 1995; Ernst et al, 2016).

A change in the physicochemical property of the membrane from lipid saturation is expected to impact the conformation, localization, or complex assembly of proteins associated with the membrane. Indeed, lipid saturation destabilizes *ERG1*-encoded squalene epoxidase, a rate-limiting enzyme in the ergosterol biosynthesis pathway. Similar to the normal turnover of Erg1 in WT, the degradation of Erg1 in *ole1-20* requires the E2 ubiquitin-conjugating enzyme Ubc7 and the E3 ubiquitin ligase Doa10. Erg1 in saturated membrane lipids might adopt different conformation or even become misfolded, rendering its recognition by the ERAD machinery. Unlike the normal turnover of Erg1 in WT, the lipid saturation–triggered degradation is independent of the E2 enzyme Ubc6, implying a different mode of Erg1 recognition by Doa10 under this condition. The partial suppression of GFP-Erg1 foci by the chemical chaperone glycerol supports the notion that Erg1 might be misfolded and aggregated. In addition, the stabilizing effect of the P469L mutation is pronounced in *ole1-20*, but not in WT, also implying an altered protein conformation under different lipid environments. K311R mutant is not stabilized in WT, which is inconsistent with the previous result (Foresti et al, 2013). The discrepancy might be attributed to different background strains used in the studies. Nevertheless, the K311R mutant is partially stabilized in *ole1-20*, suggesting that the residue K311 is involved in the saturated lipid–induced degradation, providing another evidence that Erg1 is recognized by Doa10-Ubc7 via a mechanism distinct from its normal turnover. Squalene epoxidase appears to be intrinsically sensitive to the UFA/SFA ratio of the membrane, as human SQLE is known to be stabilized by UFAs through reduced ubiquitination (Stevenson et al, 2014). The stabilization is dependent on the N-terminal 100 amino acids, which is absent in Erg1. We have also found an elevated level of Erg1 in WT supplemented with UFAs (Fig 2B). It remains to be determined whether Erg1 is also stabilized by UFAs.

It is conceivable that the membrane association domain of Erg1 might constitute the determinant for the lipid saturation–triggered degradation. Human SQLE is anchored to the ER through a re-entrant loop of ~100 amino acids at the N-terminus (Howe et al, 2015), whereas the N-terminally truncated form of SQLE, generated by proteasomal cleavage, is peripherally associated with the ER through a hairpin domain at the C-terminus (Padyana et al, 2019; Coates et al, 2021). Likewise, our differential solubilization data suggest that Erg1, structurally similar to N-terminally truncated SQLE, is not integrally associated with the membrane. Consistent with the recent biochemical study showing that Erg1 behaves as a monotopic membrane protein (Farkas et al, 2022), our MD simulation predicts that the C-terminal hairpin domain of Erg1 is partially embedded in the membrane leaflet, instead of traversing through the lipid bilayer. Although the overall conformation of Erg1 appears similar regardless of the lipid types used in the simulation, we cannot exclude the possibility that subtle conformational

variation of the protein might suffice its recognition by Ubc7-Doa10. Interestingly, local mass density along the lipid bilayer computed from the simulation shows that the amino acid P469, which is critical for lipid saturation–triggered degradation, is positioned near C9 and C10 of the acyl chain of unsaturated lipids, whereas C9 and C10 of saturated lipids shift slightly toward the center of the lipid bilayer. It raises a possibility that P469 might be accessible and become a key residue for recognition by Ubc7-Doa10 in saturated lipids, but not in a normal membrane.

The overexpression of Erg1(K311R) and Erg1(P469L) is toxic to *ole1-20*, as a result of the perturbed sterol synthesis. Erg1(P469L) is not fully functional in *ole1-20*, probably because of protein misfolding or protein clustering into foci, blocking access to its substrate. The overexpression of Erg1(P469L) greatly elevates lanosterol, but not most of the downstream sterol species in *ole1-20* (Figs 5B and S5), suggesting that the conversion of lanosterol to 4,4-dimethylcholesta-8,14,24-trienol by Erg11 might be inhibited. It remains to be determined whether the activity or stability of Erg11, an integral ER protein, might be down-regulated in *ole1-20* expressing stable Erg1 mutants. We provide two lines of evidence to support that lanosterol accumulation is toxic to *ole1-20*. First, *ole1-20* is sensitive to fluconazole, which inhibits Erg11. Second, the toxicity of Erg1(P469L) is reversed by Ro 48-8071, which inhibits the lanosterol synthase Erg7. Although lanosterol is normally present at a low level, it is detectable in intracellular membranes, but not the plasma membrane, of the budding yeast (Zinser et al, 1993). Like ergosterol, lanosterol promotes lipid acyl chain order, creating a liquid-ordered or gel-like membrane phase (Miao et al, 2002). Recent studies in bacteria show that low membrane fluidity triggers lipid phase separation and segregation of a membrane-associated protein (Gonzalez & Martin, 1996). Similarly, the increased lipid saturation in *ole1-20* may create highly packed and ordered membrane microdomains enriched for saturated lipids (Fig 8). The accumulation of lanosterol may further rigidify the membrane, leading to the aberrant membrane morphology and mislocalized membrane proteins (Fig 8).

Erg1, when overexpressed or stabilized in *doa10Δ*, forms foci in *ole1-20* at 34°C. Erg1(P469L) foci are largely colocalized with the LD marker Erg6. Our lipidomic result shows that the amount of TAG, one of the major storage lipids in LDs, is greatly reduced in *ole1* mutants (Fig S1), indicating that the Erg6-marked foci might not be LDs. Instead, these foci might represent aggregated proteins or proteins segregated to ER microdomains of distinct lipid profiles. Notably, the overexpression of either Erg1(P469L) at 27°C or WT Erg1 at 34°C does not perturb the ER morphology in *ole1-20*, and the Erg1 foci formed under these conditions are not significantly colocalized with Erg6 (Fig 4C). It is possible that ER microdomains enriched for both Erg1 and Erg6 are generated by a combination of lanosterol accumulation and increased lipid saturation in *ole1-20* at 34°C, which also perturbs the ER morphology. Whether the enrichment of Erg6 at such ER microdomains has any physiological relevance to LD biogenesis remains to be determined. The rescue of the ER morphology and the distribution of Erg1 and Nup2 by either Ro

mutations. **(D)** Cells of indicated genotypes were serially diluted and spotted on YPD plates containing DMSO or 200 ng/ml of Ro 48-8071 and incubated at RT or 32°C. **(E)** *ole1-20 doa10Δ* cells expressing indicated GFP- or mCh-tagged proteins were grown at 34°C for 4 h, in the absence or presence of Ro 48-8071 at the indicated concentration. Cells were scored for the phenotypes in the diagram and are shown as the mean ± SD from three experiments (n > 200 cells for each sample).

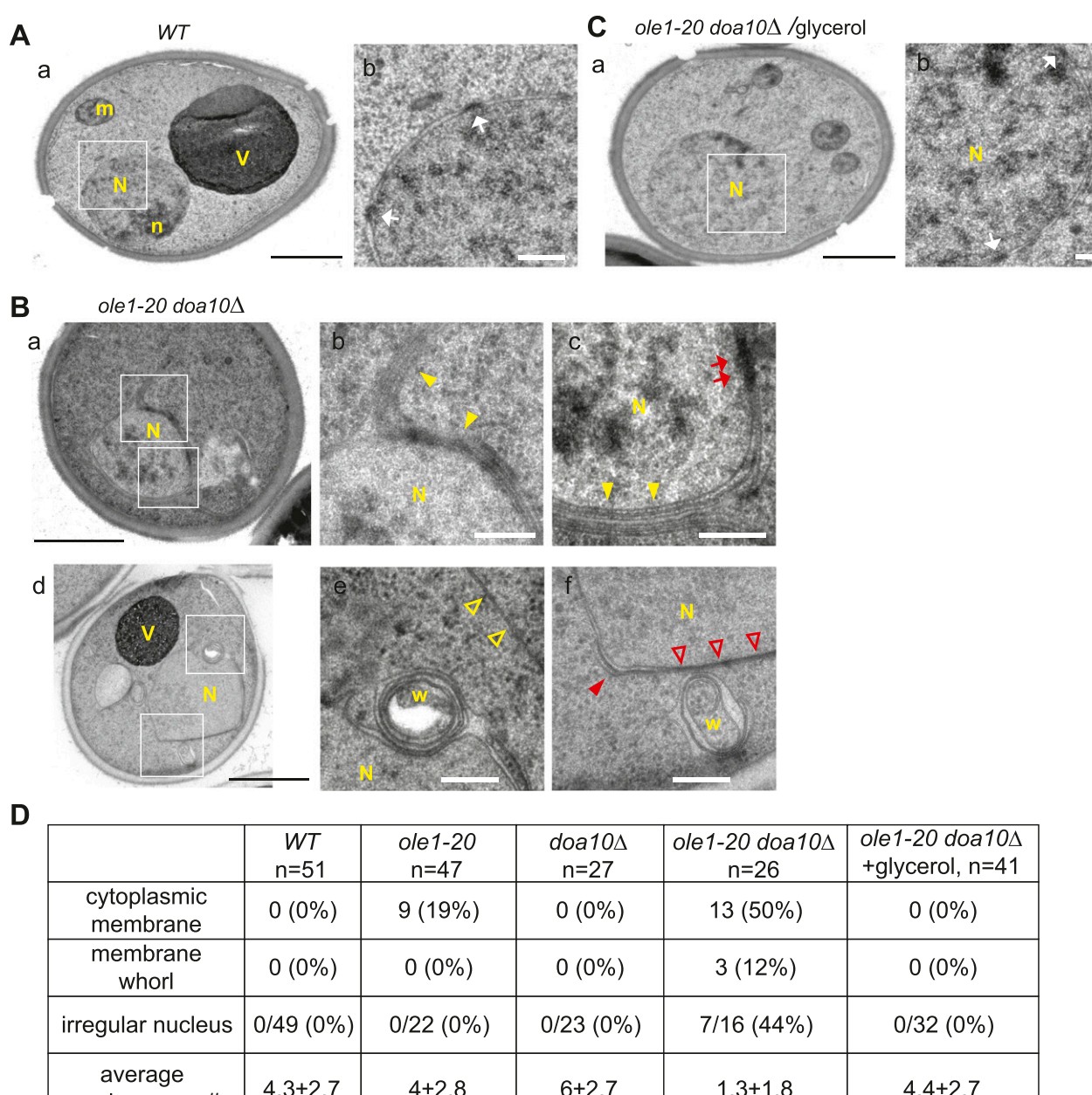

Figure 7.   Electron micrographs show aberrant membranes in *ole1-20 doa10Δ*.
**(A, B, C)** Electron micrographs of WT (A) and *ole1-20 doa10Δ* (B) grown at 34°C for 3 h. **(C)** *ole1-20 doa10Δ* with glycerol addition during incubation at 34°C. The micrographs of higher magnification for the insets in (a) and (d) are shown on the right of each panel. N, nucleus; n, nucleolus; m, mitochondria; V, vacuole; w, membrane whorl; white arrow, NPC; filled yellow arrowhead, membrane stacks; red arrow, stretch of electron-dense material in the nuclear envelope; open yellow arrowhead, cytoplasmic ER; filled red arrowhead, nuclear membrane with high curvature; open red arrowhead, juxtaposed inner and outer nuclear membrane; black scale bar, 1 μm; and white scale bar, 200 nm. **(D)** Quantification of membrane defects in the electron micrographs. The number of cells counted is indicated below the genotypes. The fraction of irregular nucleus was scored from discernible nuclei, not total cells.

48-8071 or OA supports the notion that ER phenotypes are due to the combined effect of lanosterol accumulation and increased lipid saturation in *ole1-20*. In addition, glycerol suppresses the ER phenotypes likely by preserving the conformation of Erg1 and other ER proteins in the saturated membrane lipids, as glycerol is known to stabilize the activity of enzymes and the native structure of proteins (Gekko & Timasheff, 1981). In addition, glycerol can replace water and bind to the phospholipid headgroup, increasing the area per molecule and the acyl chain disorder, thereby fluidizing the membrane (Crowe et al, 1984; Abou-Saleh et al, 2019), supporting that Erg1 conformation is sensitive to lipid packing in the membrane. Glycerol decreases squalene amount in Erg1(P469L)-overexpressing *ole1-20*, likely because of the restoration of Erg1(P469L) conformation and function or its localization in ER. The reduction in squalene by

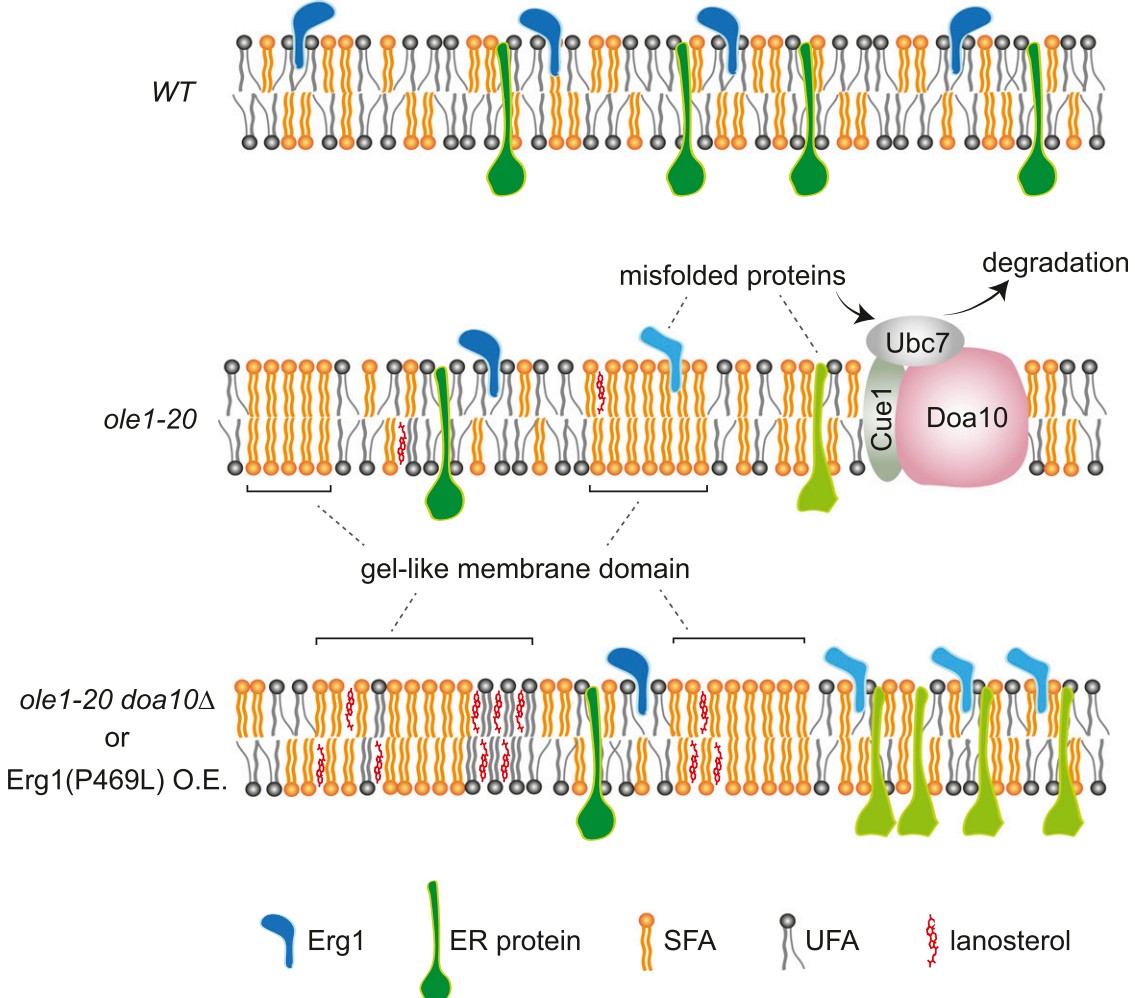

**Figure 8. Model of lipid phase separation and protein segregation in membranes of saturated lipids.**

Elevated lipid saturation in the membrane of *ole1-20* cells generates gel-phase microdomains enriched for saturated lipids. ER proteins, including Erg1, in the saturated lipids are misfolded and targeted for ERAD through the E3 enzyme Doa10 and the E2 protein Ubc7, which is recruited by Cue1 to the ER. Additional accumulation of lanosterol by *doa10Δ* or Erg1(P469L) overexpression (O.E.) expands the gel phase and causes protein segregation among different phases of the membrane, resulting in the aberrant ER morphology and mislocalized and/or dysfunctional ER proteins.

glycerol addition is not associated with a corresponding increase in lanosterol, raising the possibility that glycerol also partially alleviates the blockage of lanosterol turnover, for example, by preserving Erg11.

It is anticipated that, besides Erg1, additional membrane-associated proteins might be misfolded and targeted for degradation by Doa10 in saturated membrane lipids. The accumulation of the misfolded substrates in the *doa10Δ* deletion mutant thereby exacerbates the ER stress. The synthetic growth phenotype of *ole1-20* with *doa10Δ* and *ubc7Δ*, but not with *ubc6Δ* or *hrd1Δ*, indicates that Doa10-Ubc7 is responsible for clearing the misfolded proteins in the saturated membrane. The *ole1-20 doa10Δ* double mutant displays an aberrant ER network, similar to that in *ole1-20* over-expressing Erg1(P469L). The *ole1-20 ubc7Δ* double mutant also exhibits similar ER phenotypes, albeit to a lesser degree, suggesting that some unknown E2 enzymes might partially substitute the function of Ubc7 in clearing misfolded proteins. Electron microscopy reveals that ~50% of the *ole1-20 doa10Δ* double mutant

contained cytoplasmic ER extensions and/or ER whorls, which are reminiscent of those observed in cells under ER stress induced by UPR (Bernales et al, 2006). In addition, the double mutant displays unique nuclear membrane defects, including abnormal nuclear shape and a reduced number of NPCs. The near polygonal-shaped nuclear envelope, with low curvature in some area and high curvature in others, might be a result of membrane lipid phase separation, which generates gel-phase and liquid-disordered domains (Fig 8). The saturated lipids in *ole1-20* alone may form microdomains that cannot be distinguished by microscopy, whereas the *ole1-20 doa10Δ* double mutant may accumulate lanosterol and other lipids or proteins, which collectively rigidify the membrane and create the visible membrane domains of various curvature lines (Fig 8). It is of interest to identify other proteins targeted for degradation by lipid saturation and determine the physiological significance of the targeted degradation. Remarkably, the fluorescence microscopy result shows that Nup2, a component

of the NPC, is clustered and mislocalized to cytoplasmic ER, instead of surrounding the nucleus in *ole1-20 doa10Δ*, consistent with a reduced number of recognizable NPCs in the electron micrographs. It remains to be determined whether other NPC subunits also show the same phenotypes. As all NPC subunits display slow turnover (Hakhverdyan et al, 2021), the mislocalization of the NPC suggests that the maintenance, rather than the assembly of new NPCs, in the nuclear membrane is perturbed. It is possible that the membrane rigidity produced by an increase in saturated lipids and lanosterol disrupts the membrane curvature necessary for NPC insertion, leading to its aggregation and mislocalization. It is of interest to determine whether the mutations also affect the insertion or duplication of the spindle pole body, another macromolecule assembly, in the yeast nuclear membrane.

Studies in living bacteria have shown that a reduced membrane fluidity interferes with essential cellular processes, including chromosome replication/segregation, cytokinesis, and envelope expansion (Gohrbandt et al, 2022). We anticipate that lipid saturation also has a profound impact on the physiology of eukaryotic cells, which have more elaborate membrane systems compared with bacterial cells. By exploiting *ole1* mutants and nondegradable Erg1 proteins, it is now possible to investigate in detail how modulating membrane fluidity may impact the structure, function, and dynamics of each membrane organelle and their associated proteins.

# Materials and Methods

## Construction of yeast strains and plasmids

All strains are derivatives of W303. Gene deletions and carboxyl-terminal epitope tagging were generated by one-step PCR-mediated integration through homologous recombination at the chromosomal loci (Longtine et al, 1998). For constructing the template for *OLE1* mutagenesis, 1 kb of the *OLE1* promoter region was cloned at *Not*I and *Eco*RI, the coding region at *Eco*RI, and 500 bp downstream of the stop codon at *Eco*RI and *Xho*I in *pRS414* or *pRS416*. For an ectopic expression of 2myc- and GFP-Erg1, the promoter region (350 bp upstream of the start codon) of *ERG1* was cloned at *Sac*I and *Not*I, the 2myc or monomeric GFP sequence at *Not*I and *Bam*HI, and the coding region plus 327 bp downstream of stop codon at *Bam*HI and *Hind*III in *pRS406* or *pRS404*. The plasmids were linearized with *Stu*I and *SnaB*I for integration at *URA3* and *TRP1* locus, respectively. For *GAL*-driven GFP-Erg1, the *GAL1* promoter was cloned at *Sac*I and *Not*I, and GFP and *ERG1* were cloned as above in *pRS405*. The plasmid was linearized with *Xcm*I for integration at the *LEU2* locus. For expressing the ER marker HDEL, the *CYC1* promoter was cloned at *Sac*I-*Not*I, N-terminal 32 amino acids containing the signal sequence from *KAR2* at *Not*I-*Xba*I, 2myc at *Xba*I-*Bam*HI, mCherry or monomeric GFP at *Bam*HI-*Xma*I, and C-terminal 7 amino acids (containing the HDEL sequence) with 500 bp of 3′ region of *KAR2* at *Sal*I-*Cla*I.

## Growth of yeast

Cells were grown in YEP (1% yeast extract and 2% Bacto Peptone) containing 2% dextrose (YPD). For *GAL* induction, 2% galactose was added to cells grown in YEP containing 2% raffinose and 0.02% dextrose at an early log phase. For microscopy studies, cells were grown in a synthetic medium containing 0.67% yeast nitrogen base, complete amino acid supplement mixture (Bio101; CSM), and 2% dextrose or 2% raffinose plus 0.02% dextrose for *GAL* induction. Glycerol was added to 10% from 80% stock, along with galactose or at the time of the shift to 34°C. For fatty acid supplementation, 1% Tergitol-type NP-40 (Merck) was added to the medium to facilitate solubilization. Palmitoleic acid (Sigma-Aldrich) and OA (Sigma-Aldrich) were added directly to a liquid medium to 1 mM. Palmitic acid (Sigma-Aldrich) and stearic acid (Sigma-Aldrich) were added to 0.5 mM from 0.2 M stocks in absolute ethanol. An equal volume of ethanol was added to the medium as a control. Plates were supplemented with OA to 0.05% (~1.5 mM), fluconazole (Sigma-Aldrich) to 10 μg/ml from 10 mg/ml stock in DMSO, and Ro 48-8071 (Cayman) to 200 ng/ml from 200 μg/ml stock in DMSO. The supplements were added to the medium that has been autoclaved and cooled to 55°C before plate pouring. For microscopy studies, fluconazole and Ro 48-8071 were added to cells for 20 min before a shift to 34°C.

## Isolation of *ole1^{ts}* mutants

The *OLE1* sequence was amplified by error-prone PCR using *pRS414-OLE1* as the template in a 50 μl reaction containing 1 μM each of T3 and T7 primers, 1× Prime Taq reaction buffer (GeNet Bio), 0.2 mM dNTP, 0.1 mM MnCl$_2$, and 1 unit of Prime Taq DNA Polymerase (GeNet Bio). The PCR product was purified and transformed into *ole1Δ::Kan* strain together with *pRS416-OLE1* cut with *Eco*RI to remove the coding sequence. The transformants carrying gap-repaired *pRS416-OLE1* plasmids were selected on CSM-URA plates containing 2% glucose at room temperature (RT, ~25–27°C). The plates of ~5,000 colonies in total were replicated and placed at 37°C. The colonies that grew at RT, but not at 37°C, were individually streaked out to confirm their temperature sensitivity and their growth on selection plates containing OA. The gap-repaired *pRS416* plasmids carrying *ole1* mutants were isolated from cells, tested again for the temperature sensitivity, and sequenced. The *ole1^{ts}* sequence was isolated by cutting the plasmid with *Not*I and *Xho*I and used to transform *ole1Δ::Kan* cells. Transformants were selected for growth on YPD plates without OA and kanamycin at RT.

## Fluorescence microscopy

All images were acquired with the DeltaVision system (GE Healthcare Life Sciences), using UPlanSApo 100×/NA 1.4 oil lens (Olympus) and a sCMOS camera (pco.edge 4.2; PCO AG). Cells were first grown at 27°C to early log phase and shifted to 34°C for 3–5 h. Cells were concentrated by centrifugation and spotted onto a layer of 2% agarose made in a synthetic complete medium and covered by a coverslip. Z-stacks of 11 optical sections with a spacing of 0.5 μm were collected with the SoftWoRx software (version 7.0.0; GE Healthcare Life Sciences) and processed with the Fiji software (https://fiji.sc/). Cells were categorized for the presence of a clear perinuclear ER signal of Erg1 or ER marker proteins, or the even distribution of Nup2-GFP on the round or oval-shaped nuclei. The cell count plugin in Fiji was used to count cells in each category.

Pearson's colocalization coefficient was determined by the JACoP plugin in Fiji on deconvoluted images.

## Yeast cell lysates and Western blots

Aliquots of the cell culture were first supplemented with 20× Fix (200 mM sodium azide and 5 mg/ml BSA) to a final concentration of 1× Fix and incubated for 15 min on ice. Cells were pelleted and then incubated in 0.1 M NaOH at RT for 5 min. Cell pellets were bead-beat in 150 μl/OD$_{600}$ of urea buffer (8 M urea, 4% SDS, 1 mM EDTA, 5% $\beta$-mercaptoethanol, and 100 mM Tris, pH 6.8) and incubated at 55°C for 15 min. Samples were subjected to SDS–PAGE, followed by Western blotting using the following antibodies: anti-Erg1 and anti-mCherry (provided by Chao-Wen Wang, Institute of Plant and Microbial Biology, Academia Sinica), anti-myc (9E10; Covance), anti-actin (MAB1501; Chemicon), and anti-Pgk1 (Abcam).

## Differentiation solubilization

Aliquots of 20–30 OD$_{600}$ of cells were collected and subjected to spheroplasting with zymolyase (Zymo Research). The spheroplasts were lysed by Dounce homogenization for 30 strokes in TNE buffer (50 mM Tris, pH 7.2, 75 mM NaCl, 5 mM EDTA, 1 mM PMSF, and 10 μg/ml each of leupeptin, pepstatin, and chymostatin [LPC]). The samples were centrifuged at 300$g$ for 5 min to remove cell debris. The crude cell lysate in the supernatant was then divided into four tubes and centrifuged at 16,100$g$ for 20 min. The resulting microsome pellets were resuspended in TNE, detergent buffer (10 mM Tris, pH 7.2, and 1% Triton X-100), alkaline buffer (0.1 M Na$_2$CO$_3$), or high-salt buffer (10 mM Tris, pH 7.2, and 0.5 M NaCl), all containing LPC and 1 mM PMSF. The samples were incubated at 4°C for 30 min and then centrifuged at 16,100$g$ for 20 min. The supernatants were supplemented with Triton X-100 and NaCl to a final concentration of 1% and 0.5 M, respectively. The pellets were resuspended in the same volumes of TNE containing 1% Triton X-100, 0.5 M NaCl, LPC, and 1 mM PMSF. After adding SDS–PAGE sample buffer, the samples were then denatured by heating at 65°C for 20 min and subjected to SDS–PAGE and Western blotting.

## Lipid extraction of glycerolipids and glycerophospholipids

Cells were grown to an early log phase at 27°C and then shifted to 34°C for 5 h to OD$_{600}$ ~1. Lipid extraction was modified from Folch's method and described elsewhere (Folch et al, 1957; Hsu et al, 2017). Briefly, 40 OD$_{600}$ of cells were resuspended in 10 mM of NaN$_3$ for 15 min on ice, and then pelleted in four tubes and stored at –80°C until use. Cell pellets were mixed with ~100 μl of glass beads (Biospec Products), 330 μl of methanol (J.T.Baker, Thermo Fisher Scientific), 5 μl of internal standard cocktail containing 1 mg/ml of hepta-decanoic acid (C17:0; Sigma-Aldrich), and 333 μg/ml of d5-TG (17:0/17:1/17:0) (Avanti). After bead-beating at RT for 10 min, samples were mixed with 660 μl of chloroform (ACROS) and then centrifuged at 10,000$g$ for 10 min to remove debris. The supernatant was transferred to new tubes and mixed with 0.9% sodium chloride to a final concentration of 0.14% to induce phase separation, followed by centrifugation at 10,000$g$ for 10 min at RT. The upper aqueous phase was discarded, and the organic phase, containing lipids, was dried

with SpeedVac (Thermo Fisher Scientific). Lipid extracts were dissolved in 150 μl of chloroform/methanol (2:1) solution before LC-MS analysis.

## LC-MS analysis

For analyzing glycerolipids and glycerophospholipids, a linear ion trap–Orbitrap mass spectrometer (Orbitrap Elite; Thermo Fisher Scientific) was used and coupled online with an ultra-high-performance LC system (ACQUITY UPLC; Waters). Chromatographic separation was performed on a reverse-phase column (CSH C18, 1.8 μm, 2.1 × 100 mm; Waters) maintained at 65°C using mobile phase A (10 mM ammonium acetate, pH 5.0, and 40% acetonitrile) and mobile phase B (10 mM ammonium acetate, pH 5.0, and 10% acetonitrile in isopropyl alcohol) in a gradient program (0–10 min: 40–99.9% B; 10–12 min: 99.9% B; 12–13 min: 99.9–40% B; and 13–14 min: 40% B; linear gradient for all) with a flow rate of 500 μl/min. The mass spectrometer was operated in positive- and negative-ion modes and set to one full Fourier transform MS scan (m/z = 100–1,200, resolution = 30,000). The mass-to-charge ratio (m/z) of TAG and DAG species was searched as [M + NH$_4$]$^+$ ions, PC and lysoPC species as [M + H]$^+$ ions, and other classes of phospholipid, lyso-phospholipid, and free fatty acid species as [M-H]$^-$ ions. The lipids were quantified with the Xcalibur software (Thermo Fisher Scientific) and analyzed with Excel (Microsoft). Data were normalized by either heptadecanoic acid (negative-ion mode) or d5-TG (positive-ion mode).

## Fatty acid methylation and GC-MS analysis

The methylation of fatty acid was carried out as described (Morrison & Smith, 1964). Dried lipids extracted from 20 OD$_{600}$ of cells were resuspended in 230 μl of 14% bromotrifluoride in methanol (Sigma-Aldrich). The mixture was heated to 95°C for 10 min and then cooled down to RT for 10 min. To increase the solubility of TAG, 198 μl of benzene (Alfa Aesar) was added. Samples were incubated at 95°C for 30 min and then cooled down to RT. Phase separation was performed by mixing with 690 μl of petroleum ether (Sigma-Aldrich) and 230 μl of ddH$_2$O, followed by centrifugation at 9,600$g$ for 10 min. Lipid fraction (upper layer) was dried using Savant SPD111V SpeedVac equipped with a RVT400 condensation tank and a M22NT pump (Thermo Fisher Scientific). Samples were resuspended in 150 μl of chloroform/methanol (2:1), 1 μl of which was subjected to a 7890A GC system (Agilent) using a DB-5MS column and analyzed by a 5975C mass spectrometer (Agilent), with the injection temperature at 250°C, the interface at 280°C, and the ion source at 230°C. Helium was used as a carrier gas with a constant flow rate of 1.1 ml/min. The column temperature was started at 80°C for 1 min, ramped to 231°C at 7°C/min, to 240°C at 3°C/min, and to 251°C at 6°C/min, and held for 9 min. Finally, the temperature was ramped to 282°C at 9°C/min and held for 3 min, in a total run of 45 min. Fatty acid methyl esters were identified based on the NIST mass spectral library (https://chemdata.nist.gov). Full-scan mass spectra were recorded in the 27–540 m/z range (0.357 s per scan). Quantification was done by integration of the area of the total ion chromatogram in the peak of the compound. All values were normalized by internal standard (C17:0). The Excel software was used for statistical operation.

## Total sterol extraction and GC-MS analysis

Sterols were extracted as described (Guan et al, 2009). Forty $OD_{600}$ of cells were collected in a glass tube (Sigma-Aldrich) and incubated in 20 ml of 10 mM $NaN_3$ for 15 min. Cells were pelleted down and resuspended in 750 μl of methanol, 500 μl of 60% KOH, 20 μl of 2 mg/ml cholesterol (served as an internal standard) (Sigma-Aldrich), and ~300 μl of glass beads. Tubes were filled with argon before bead-beating for 5 min and heated to 85°C in a water bath for 2 h for alkaline hydrolysis. After cooling down to RT, sterol and sterol ester were extracted 4 times each with 1.2 ml petroleum ether (34491; Honeywell) and dried in a 1.5-ml microtube by SpeedVac (Thermo Fisher Scientific) at RT. For derivatization, total sterol extracts were resuspended in 50 μl of pyridine (Alfa Aesar) and 50 μl of BSTFA (Sigma-Aldrich). The mixture was incubated in a 60°C water bath for 30 min, diluted by ethyl acetate (Sigma-Aldrich), and subjected to GC-MS analysis immediately as described above. The column temperature was started at 100°C for 1 min, ramped to 250°C at 10°C/min, to 289°C at 3°C/min, and to 310°C at 1.5°C/min, and held for 3 min, in a total run of 46 min. The trimethylsilyl ether derivatives of sterols were identified based on the NIST mass spectral library. Full-scan mass spectra were recorded in the 50–650 m/z range (0.213 s per scan). Quantification was done by integration of the area of the total ion chromatogram in the identified peak. The value was normalized by an internal standard (cholesterol). Statistical analysis was performed with Excel.

## MD simulation

Erg1 structure was predicted by AlphaFold2. The MD simulation was performed with GROMACS (v2020.5 or later) (https://www.gromacs.org/) using the CHARMM36m force field and the TIP3P water model. The input scripts for GROMACS, including Erg1 protein, lipid bilayer, and water molecule assembly, were generated by Charmm-GUI (https://www.charmm-gui.org/). The initial orientation of Erg1 was predicted using the PPM server (https://opm.phar.umich.edu/ppm_server2_cgopm). The box system contains ~426 lipid molecules and was charge-neutralized by adding $K^+$ and $Cl^-$ counter-ions to 150 mM for energy minimization. During the NPAT (constant normal pressure and lateral surface area of membranes and constant temperature) equilibration, the system was held at 303.15 K with the v-rescale method and 1 bar with the Berendsen method. During production simulations, the Parrinello–Rahman barostat method and the Nose–Hoover thermostat were used to maintain the pressure and the temperature, respectively. The time step for production was set to 0.002 ps. Periodic boundary conditions were used in all directions. All hydrogen bonds in energy minimization, NPAT equilibration, and MD production steps were constrained with the LINCS algorithm. A 500-ns simulation of the MD production was carried out. The trajectories were visualized using the PyMOL software. The local mass density was computed along the bilayer normal using the "gmx density" command of GROMACS and visualized by Excel.

## Electron microscopy

Cells were pelleted and transferred into gold planchette (100 μm deep; Leica) for immediate cryofixation by high-pressure freezing (High-Pressure Freezer EM HPM100; Leica), followed by freeze substitution and low-temperature embedding (EM AFS2; Leica). Freeze substitution was performed in 2% osmium tetroxide and 0.1% uranyl acetate in acetone (Electron Microscopy Sciences; EMS) at −85°C for 66 h, −50°C for 10 h, −20°C for 10 h, 0°C for 6 h, and RT for 2 h, with 5°C/h increase between steps. Samples were separated from the gold planchette by pipetting at RT, then washed in acetone and subjected to infiltration by an increasing gradient of 5, 10, 25, 50, and 75, and washed three times with pure Embed 812 resin. The infiltration took at least 2 h for each concentration, except that the final infiltration with pure resin was done overnight. After polymerization at 70°C for 48 h, the blocks were cut into 70- to 90-nm ultrathin sections on the ultramicrotome (EM UC6; Leica) with a diamond knife (Ultra 45°; Diatome). Sections were placed on 0.4% Formvar-coated slot grids (EMS) and imaged at 80–100 kV on a transmission electron microscope (Tecnai G2 Spirit TWIN; FEI Thermo Fisher Scientific) using a digital CCD camera (832 Orius SC1000B; Gatan). To enhance the contrast, the sections on the slot were further stained with 2% gallic acid plus 1% of tannic acid in 0.1 M cacodylate buffer for 5 min, followed by three washes with $ddH_2O$ for 1 min. A second stain was done with 4% aqueous uranyl acetate for 5 min, followed by three washes with $ddH_2O$ for 1 min. At the end, the samples were stained with Reynold's lead citrate for 5 min, followed by four washes with $ddH_2O$. The images were processed with the Fiji software.

## Tetrad analysis

The deletion mutants of ERAD components marked by the kanamycin-resistant gene were crossed to *ole1-20* of the opposite mating type. The diploid heterozygous mutants were then sporulated for 2 d in the sporulation medium (1% potassium acetate, 0.1% yeast extract, and 0.05% dextrose), followed by brief treatment with zymolyase to remove the ascus. The cells in each tetrad were manually picked under a microscope and placed on YPD plates. After growing at RT, the emerging colonies were streaked out on YPD plates and tested for their genotypes by replica plating onto YPD containing kanamycin at RT or onto YPD at 34°C.

## SILAC proteomics

For SILAC labeling, *lys2Δ* and *ole1-20 lys2Δ*, which are auxotrophic for lysine, were grown in 5 ml of synthetic complete media containing 1 mM L-lysine (light) for *lys2Δ* or 1 mM L-lysine [$^{13}C_6^{15}N_2$] (heavy) for *ole1-20 lys2Δ* overnight. Cultures were then diluted to $OD_{600}$ ~0.15 in 15 ml of fresh medium of the same composition. Cells were grown to $OD_{600}$ ~0.7, and 8.5 $OD_{600}$ of each culture were mixed, harvested by centrifugation, and incubated with 10 mM sodium azide for 15 min on ice. The samples were then treated with zymolyase to generate spheroplasts. Total cell extracts were prepared by bead-beating the spheroplasts in lysis buffer (8 M urea, 100 mM Tris–Cl, pH 7.8, 4% SDS, 1 mM EDTA, and 5% β-mercaptoethanol) and heated at 60°C for 15 min. After removing cell debris by centrifugation at 16,100*g*, the cell extracts were reduced with 10 mM DTT and then alkylated with 25 mM iodoacetamide. Urea concentration was then reduced to 4 M and the pH adjusted with 50 mM Tris–HCl (pH 9). Samples were digested with

Lys-C in a 1:10 enzyme–protein ratio at 37°C for 16 h, and then desalted using an Oasis Plus HLB cartridge (Waters). Digests were fractionated using electrostatic repulsion–hydrophilic interaction chromatography. A C18 capillary column (75 μm × 250 mm, 1.7 μm, BEH130; Waters) was used to separate peptides. The nano-UPLC system (nanoACQUITY; Waters) coupled to an LTQ Orbitrap Velos hybrid mass spectrometer (Thermo Fisher Scientific) was used for protein identification and analysis. Data analysis was performed using the Proteome Discoverer software suite (v1.4; Thermo Fisher Scientific), and the SEQUEST and Mascot (v2.3; Matrix Science) search engines were used for peptide identification. Data were searched against an in-house–generated database containing all proteins in the *Saccharomyces* Genome Database plus the most common contaminants. The identified peptides were filtered using an FDR lower than 1%. Peptide areas were used to calculate the heavy–light ratios.

## Supplementary Information

## Acknowledgements

We thank Chao-Wen Wang (Institute of Plant and Microbial Biology, Academia Sinica, Taiwan) for providing anti-Erg1 and anti-mCherry antisera, scientific input, and help in lipidomic analysis; Hsin-Nan Lin and Ching-Chen Liu (Bioinformatics Core Facility, Institute of Molecular Biology, Academia Sinica, Taiwan) for help in molecular dynamics simulation; and Tzu-Han Hsu (Imaging Core Facility, Institute of Molecular Biology, Academia Sinica, Taiwan) for help in electron microscopy. The research was supported by an intramural fund from Academia Sinica, Taiwan.

### Author Contributions

L-J Huang: resources, data curation, software, formal analysis, validation, investigation, visualization, and methodology.
R-H Chen: conceptualization, resources, data curation, formal analysis, supervision, funding acquisition, validation, investigation, visualization, methodology, project administration, and writing—original draft, review, and editing.

### Conflict of Interest Statement

The authors declare that they have no conflict of interest.

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
