## [Reviewer comments · Life Science Alliance]

Life Science Alliance

Lipid saturation induces degradation of squalene epoxidase for sterol homeostasis and cell survival

Rey-Huei Chen and Leng-Jie Huang

DOI: <https://doi.org/10.26508/lsa.202201612>

Corresponding author(s): Rey-Huei Chen, Institute of Molecular Biology, Academia Sinica

Review Timeline:

Submission Date:	2022-07-19
Editorial Decision:	2022-08-11
Revision Received:	2022-10-14
Editorial Decision:	2022-10-24
Revision Received:	2022-10-24
Accepted:	2022-10-25

Scientific Editor: Novella Guidi

Transaction Report:

August 11, 2022

Re: Life Science Alliance manuscript #LSA-2022-01612-T

Dr. Rey-Huei Chen
Institute of Molecular Biology, Academia Sinica
Institute of Molecular Biology
128 Sec. 2, Academia Rd.
Taipei 11529
Taiwan

Dear Dr. Chen,

Thank you for submitting your manuscript entitled "Lipid saturation induces degradation of squalene epoxidase for sterol homeostasis and cell survival" to Life Science Alliance. The manuscript was assessed by expert reviewers, whose comments are appended to this letter. We invite you to submit a revised manuscript addressing the Reviewer comments.

Thank you for this interesting contribution to Life Science Alliance. We are looking forward to receiving your revised manuscript.

Sincerely,

B. MANUSCRIPT ORGANIZATION AND FORMATTING:

Reviewer #1 (Comments to the Authors (Required)):

In this manuscript Huang and Chen explore the effects of phospholipid saturation on stability of a model membrane protein in yeast, Erg1. To control saturation, they generated temperature-sensitive mutants in the desaturase Ole1, and they obtained a mutant that accumulates saturated fatty acids at high temperature that is toxic to cell survival. Authors found that Erg1, a protein in the ergosterol biosynthetic pathway, is unstable stable in this ole1ts mutant at high temperature. Supplementation with oleic acid (OA) yields stable Erg1. They found that Doa1 and Ubc7, proteins of the ERAD complex, is involved in the degradation of Erg1 upon lipid saturation. The authors then identified Erg1 mutants resistant to degradation upon lipid saturation. They determined cell localization of Erg1 (normally in the ER) and found that it forms foci upon lipid saturation, and that ER morphology was aberrant. The toxic effects of saturation on cells were attributable to these changes in ER morphology, which, in turn, was caused by accumulation of lanosterol. Finally, they demonstrated that addition of glycerol resulted in normal ER morphology and prevention of protein misfolding, presumably due to a chaperone-like effect of this carbon source.

These data are significant in that they establish a new model for tightly temporally regulating phospholipid saturation in yeast, and they demonstrate that folding and stability of membrane proteins, as well as organellar morphology, are exquisitely dependent on this membrane parameter. The paper is easy to read, and the data are appropriately interpreted.

However, there are issues with the manuscript:

- (1) It is not clear whether glycerol, used as a source of carbon, can directly affect the expression level of the Erg6 mutants under the Gal promoter. Does glycerol change the morphology of the ER?
- (2) The authors showed that the stable Erg1 mutants are toxic for the ole1-20 strain (Fig. 5A), and this is rescued by OA supplementation. Does OA change the localization of Erg1 mutants and wt? What is the ER morphology in the ole1-20 strain during OA supplementation? The authors might observe GFP-Erg1 and mCherry-HDEL-expressed strain in the presence of OA.
- (3) The authors use fluconazole and Ro48-8071 to determine lanosterol toxicity. What happens to cells if they are directly fed lanosterol? What is the status of ER morphology and Erg1 and Erg1 variant localization after fluconazole and R048-8071, or exposure to lanosterol?
- (4) In Figure 6 the authors tested the status of ER morphology in the ole1-20doa10 strain in the presence and absence of glycerol as chemical chaperone. However, they did not determine in parallel this effect in a ole1-20ubc7Δ mutant. This is important to determine the relative roles of these two degradative proteins in ER morphology.

Minor comments:

- (5) Be consistent with spaces (a) between temperature and {degree sign}C (see Fig 2 legend); and (b) between number and "μm" in scale bars (see Fig 6 legend).
- (6) In some experiments authors used 34oC and some experiments with 32oC. Is there a justification?
- (7) Similarly, "RT" is sometimes used, and 27{degree sign}C is also used. Is there a difference? Please keep uniform.

Reviewer #2 (Comments to the Authors (Required)):

This study investigated phenotypes associated with OLE1 loss and increased lipid saturation in budding yeast, and discovered accelerated degradation of squalene epoxidase (Erg1) under these conditions. Huang & Chen show that this degradation is mediated by the ERAD E3 ubiquitin ligase Doa10 and depends on a conserved proline residue in the membrane-associated domain of Erg1. Interesting, combined OLE1 loss and overexpression of degradation-resistant Erg1 mutants dramatically disrupts ER morphology and impairs cell growth due to the toxic accumulation of lanosterol, suggesting that lipid saturation-induced degradation of Erg1 is necessary to avert this accumulation. Finally, the authors show that loss of Doa10 mimics the effect of Erg1 mutants in causing cell-wide, lanosterol-dependent membrane disruption and toxicity in cells lacking OLE1. Taken together, the study presents a model in which Doa10 is required to clear misfolded proteins from saturated lipid-rich membranes, which in the case of Erg1 prevents aberrant lanosterol accumulation and further ER dysfunction. This finding reveals a new element in the regulation of lipid homeostasis and is likely to be of interest to the journal readership.

Overall, the authors have used a wide range of assays and logical sequence of experiments to support their conclusions. It is convincingly shown that lipid saturation controls Erg1 degradation and that this is required for normal ER function; however, other aspects of the model (e.g., lanosterol accumulation) would benefit from further elaboration.

Major comments

1. The concept of the effect of unsaturated vs saturated fatty acids stabilizing the mammalian equivalent of Erg1 was introduced by Stevenson et al. (2014 *Biochem J*). This paper was overlooked and should be appropriately discussed here in the context of the present yeast work.
2. Fig. 3C: Erg1 K311R is degraded at the same rate as the WT protein, which seems to conflict with the findings of Foresti et al. (2013 *eLife*). Furthermore, its degradation in *ole1-20* cells appears blunted, suggesting that Lys-311 is involved in the saturated lipid-induced degradation of Erg1. Can the authors please comment on these results in the manuscript?
3. Fig. 2D: the predicted Erg1 conformational change in completely saturated vs. completely unsaturated lipid bilayers is relatively subtle. What is the predicted conformation in a bilayer with a PSPC:YOPC ratio that mimics a 'typical' proportion of saturated and unsaturated lipids? Is a change still apparent?
4. Fig. 5B: the central role of lanosterol accumulation in the proposed mechanism of ER dysfunction is interesting but warrants further clarification. Fig. EV3 (which is not mentioned in the results section) indicates that the levels of downstream ergosterol synthesis intermediates are generally reduced in *ole1-20* cells, pointing to a pathway blockage at lanosterol. Why does this occur? One possibility is that lipid saturation also impairs the protein stability or activity of lanosterol demethylase (Erg11), which warrants testing given that the human homologue of Erg11 (LDM) is targeted by the homologue of Doa10 (MARCHF6) (Scott et al. 2021 *Biochem J*).
5. Fig. 5B: squalene conversion is restored by the addition of glycerol in Erg1 P469L-expressing *ole1-20* cells; however, there is no corresponding increase in lanosterol levels. Can the authors please comment on this result? As a point of comparison, this experiment would benefit from including cells expressing WT Erg1 (where the ER is intact) to confirm that this indeed averts or blunts lanosterol accumulation.
6. Fig. 5D: does manipulating lanosterol levels also restore ER morphology? The authors have only considered cell growth. If so, it would help to support their final model.

Minor comments

1. The statement "both mammalian SQLE and Erg1 are destabilized in response to an elevated level of downstream intermediate lanosterol" is incorrect - degradation of mammalian SQLE is unaffected by lanosterol (Gill et al. 2011 *Cell Metab.*).
2. On page 9, it is stated that "Erg1 structure predicted by AlphaFold2 was similar to SQLE (Fig 3A)." However, no comparison between the two proteins is depicted.
3. Please include methods for the scoring of imaged cells (e.g., Fig. 4B) and tetrad analysis (Fig. 6C).
4. To improve clarity, labels should be added to the icons in Figs. 4A and 6B to explicitly describe the phenotype.
5. The schematic in Fig. 8 is a useful accompaniment to the discussion, but would benefit from inclusion of Erg1, Doa10 and Ubc7 to better integrate the findings of the entire manuscript. The legend does not outline the possible consequences of protein segregation into different membrane phases - presumably this is the likely cause of the membrane dysfunction?
6. Please ensure that figure legends contain full information about temperature(s) at which experiments were performed and any medium supplements (e.g., oleic acid, galactose).
7. Please define the acronyms "TAG" and "NPC" in the main text.
8. Please correct typos in the main text and figures, e.g., "lesser extend" (page 11), "microscopy. Whereas" (page 19), "GFP-P469L" (Fig. 4C), "deleted for" (Fig. 2D legend), "Student's" (Fig. 4C legend).

Reviewer #3 (Comments to the Authors (Required)):

In their study, Huang and Chen investigate the cellular consequences of manipulating glycerophospholipid saturation and thus membrane properties in *S. cerevisiae*. They first generate temperature-sensitive mutant alleles of the single yeast lipid desaturase *Ole1*, an approach motivated by the fact that the *OLE1* gene is essential. These mutant alleles enable them to discover that increasing lipid saturation accelerates ER-associated degradation of the squalene epoxidase *Erg1* and thereby impacts the ergosterol biosynthesis pathway. The physiological importance of accelerated *Erg1* degradation is revealed when the authors raise the levels of *Erg1* under conditions of increased lipid saturation and find a toxic accumulation of the ergosterol synthesis intermediate lanosterol. Lanosterol accumulation under these conditions disrupts ER structure and impairs cells survival, likely because of further increasing lipid order in the ER membrane.

These findings uncover an unanticipated link between glycerophospholipid saturation, ERAD and sterol synthesis. It advances our understanding of lipid homeostasis and the importance of balancing the abundance of saturated and unsaturated glycerophospholipids and sterols. While the exact mechanisms of lipid saturation sensing and subsequent ERAD of *Erg1* remains to be elucidated, this is a well-done paper. It is technically convincing throughout and is also nicely written, with a very logical and informative introduction. There are only minor issues to be resolved (see below). The only experimental revision I recommend is to add a true ER transmembrane protein as a control for the subcellular fractionation experiment shown in Figure 3F, which would be a better reference point than the ER-luminal GFP-HDEL the authors included so far. This would strengthen the claim that *Erg1* has a somewhat unusual mode of membrane association. Other than that, the manuscript is suitable for publication very much as is and I congratulate the authors on their fine piece of research.

Minor issues:

P5: When explaining that HMGR becomes misfolded, please cite one of the more recent papers from Randy Hampton, i.e. either Wangeline and Hampton, JBC 2018 or Wangeline and Hampton, JBC 2021.

P6: "... lipotoxicity-related diseases (Schaffer, 2016)." Please give examples of the diseases you are referring to.

P7: "The *ole1Δ* deletion mutant is lethal". The mutant is not lethal but dead. Hence, more accurate ways of phrasing this would be "The *ole1Δ* deletion is lethal" or "The *ole1Δ* deletion mutation is lethal."

P7: For the benefit of orienting the reader, please begin with a statement why *OLE1* ts alleles were made.

P7, Figure 1B: In the figure it says "recovered" instead of recovered.

P7, Figure 1D: Please add information to the figure itself that growth is at 34 degrees. This piece of information is in the text and in the figure legend but it would help the reader to also see it in the panel.

P8: The authors mention that SILAC was done to find proteins with differential abundance in WT versus *ole1-20* cells but the data are not provided. Please either include the data or add a comment that or why they are not shown.

P9: "... showed that *Erg1* situated on the surface ...". Please edit to: "... *Erg1* is situated on ..."

P10, Figure 3F: Please use a transmembrane protein as a reference rather than GFP-HDEL. The GFP-HDEL only shows that carbonate and high salt did not disrupt the ER membrane. However, to illustrate that *Erg1* is not a regular integral membrane protein, it is necessary to show that the solubility of a true transmembrane protein is unaffected by carbonate treatment.

P11: "... and to a lesser extent at 27 ...", should be: "... and, to a lesser extent, at 27 ..."

P11: Could the foci of *Erg1*(P469L) in *ole1-20* cells that co-localize with *Erg6* but are not lipid droplets (Figure 4) represent ERAD foci (Albert, ... Engel, PNAS 2020)? Please comment in the discussion if you think that this is a reasonable possibility. This is entirely optional, though, the authors do not have to include this idea if they do not think that it is helpful.

P15: "... and for the membrane microdomain of specific functions." Not clear what membrane microdomain is meant here. If you refer to membrane microdomains generally, then edit to: "... for membrane microdomains of specific functions."

Below is the list of new data included in the manuscript, followed by our point-by-point response to reviewers' comments.

New data:

3A, structure of human SQLE protein

3F, differentiation solubilization using integral membrane protein Sec61 as control

5C, sterol analysis of *ole1-20* cells expressing wild-type Erg1 and Erg1 P469L

5G, Ro 48-8071 addition suppressed ER phenotypes in *ole1-20* overexpressing stable Erg1

6E, Ro 48-8071 addition suppressed ER phenotypes in *ole1-20 doa10Δ* double mutant

S2, Erg1 identified in SILAC proteomics

S3, molecular dynamics simulation of Erg1 in lipids mimicking the membranes of WT and *ole1-20*

S6, images of aberrant ER in *ole1-20 ubc7Δ* double mutant

S7, suppression of ER phenotypes by oleic acid treatment

Reviewer #1

(1) It is not clear whether glycerol, used as a source of carbon, can directly affect the expression level of the Erg6 mutants under the Gal promoter. Does glycerol change the morphology of the ER?

The addition of glycerol to cells grown in medium containing glucose or galactose does not normally change ER morphology. Whereas glycerol can restore ER morphology in *ole1* mutant expressing stable Erg1. The effect was not due to suppression of Erg1 expression from the *GAL* promoter. In fact, glycerol addition slightly increases Erg1 abundance, likely due to a stabilizing effect. In addition, the effect of glycerol is independent of its metabolism, because ER morphology is still restored with the deletion of Gut1 that converts glycerol to glycerol-3-phosphate, the first step of glycerol metabolism. We did not include the additional control data in order to save space.

(2) The authors showed that the stable Erg1 mutants are toxic for the *ole1-20* strain (Fig. 5A), and this is rescued by OA supplementation. Does OA change the localization of Erg1 mutants and wt? What is the ER morphology in the *ole1-20* strain during OA supplementation? The authors might observe GFP-Erg1 and mCherry-HDEL-expressed strain in the presence of OA.

We have performed the experiments as suggested by the reviewer. As expected, OA addition restored the ER morphology in *ole1-20* overexpressing stable Erg1 mutant and in *ole1-20 doa10* double mutant, supporting our notion that the combination of lipid saturation and lanosterol results in aberrant ER morphology. The data is now shown in Fig S7.

(3) The authors use fluconazole and Ro48-8071 to determine lanosterol toxicity. What happens to cells if they are directly fed lanosterol? What is the status of ER morphology and Erg1 and Erg1 variant localization after fluconazole and R048-8071, or exposure to lanosterol?

For the effect of Ro 48-8071, we now show that Erg1 distribution and ER morphology are indeed restored by Ro 48-8071 in a dose-dependent manner in *ole1-20* overexpressing stable Erg1 (Fig. 5G) and in *ole1-20 doa10* double mutant (Fig. 6E). For direct lanosterol addition, we did not see perturbed ER morphology in *ole1-20* after incubation with lanosterol at a range of concentrations for 3-5 hr. However, it is unclear whether exogenous lanosterol can be incorporated into the ER, or simply retained in the plasma membrane. It's also unclear whether *ole1-20*, with a relatively normal level of ergosterol, can uptake lanosterol. As to fluconazole, we found that ER morphology was not affected by adding fluconazole to *ole1-20* cells for 3-5 hr at 34°C. We think that the short drug treatment for imaging might not reflect the fluconazole sensitivity of *ole1-20* grown on plates for 2-3 days. Alternatively, the toxicity from fluconazole might be due to some unknown effect of lanosterol accumulation in *ole1-20*, which might not be directly associated with aberrant ER morphology.

(4) In Figure 6 the authors tested the status of ER morphology in the *ole1-20doa10Δ* strain in the presence and absence of glycerol as chemical chaperone. However, they did not determine in parallel this effect in a *ole1-20ubc7Δ* mutant. This is important to determine the relative roles of these two degradative proteins in ER morphology.

We now show that *ole1-20 ubc7Δ* exhibits aberrant ER morphology and clustering of Erg1 and Nup2, albeit to a lesser degree than does *ole1-20 doa10Δ* (Fig. S). The result supports that Doa10-Ubc7 complex acts together to clear misfolded proteins in *ole1-20*, although some unknown E2 enzyme might partially substitute the function of Ubc7.

(5) Be consistent with spaces (a) between temperature and {degree sign}C (see Fig 2 legend); and (b) between number and "μm" in scale bars (see Fig 6 legend).

We have fixed the spaces accordingly to be consistent throughout the text.

(6) In some experiments authors used 34oC and some experiments with 32oC. Is there a justification?

The plate assay in Figure 6D was done at 32°C, which is a semi-permissive temperature for *ole1-20*, but nonpermissive for *ole1-20 doa10Δ* and *ole1-20 ubc7Δ* double mutants. The addition of Ro 48-8071 rescued the lethality of the double mutants at 32°C. This experiment was done at 32°C in order to show the difference in temperature sensitivity between *ole1-20* and the double mutants.

(7) Similarly, "RT" is sometimes used, and 27{degree sign}C is also used. Is there a difference? Please keep uniform.

Due to limited availability of the incubator, we usually placed the plates on the bench for the permissive temperature. "RT" is used, because the temperature in the lab is maintained at 25-27°C, as described in the main text and the materials and methods.

Reviewer #2

Major comments

1. The concept of the effect of unsaturated vs saturated fatty acids stabilizing the mammalian equivalent

of Erg1 was introduced by Stevenson et al. (2014 Biochem J). This paper was overlooked and should be appropriately discussed here in the context of the present yeast work.

We have now discussed the findings from Stevenson et al. (2014 Biochem J) on the stabilization of mammalian equivalent of Erg1 by unsaturated fatty acids in the end of the third paragraph of the discussion section.

2. Fig. 3C: Erg1 K311R is degraded at the same rate as the WT protein, which seems to conflict with the findings of Foresti et al. (2013 eLife). Furthermore, its degradation in *ole1-20* cells appears blunted, suggesting that Lys-311 is involved in the saturated lipid-induced degradation of Erg1. Can the authors please comment on these results in the manuscript?

Erg1 K311R is not stabilized in our wild-type cells, which is inconsistent with previous findings of Foresti et al. (2013 eLife). The only explanation we have for this discrepancy is different background strains used in the studies (W303 vs BY4741). K311R mutant is partially stabilized in *ole1-20*, suggesting the K311 is involved in the degradation triggered by lipid saturation. The result also supports that Erg1 might adopt altered conformation and be targeted to degradation via distinct mechanism in saturated lipid membranes. We have now added this information in the third paragraph of the discussion section.

3. Fig. 2D: the predicted Erg1 conformational change in completely saturated vs. completely unsaturated lipid bilayers is relatively subtle. What is the predicted conformation in a bilayer with a PSPC:YOPC ratio that mimics a 'typical' proportion of saturated and unsaturated lipids? Is a change still apparent?

We have now performed molecular dynamics simulation of Erg1 in a mix of saturated and unsaturated lipids that mimics the membranes of WT and *ole1-20* cells. The result still did not show apparent conformational difference (Fig. S3).

4. Fig. 5B: the central role of lanosterol accumulation in the proposed mechanism of ER dysfunction is interesting but warrants further clarification. Fig. EV3 (which is not mentioned in the results section) indicates that the levels of downstream ergosterol synthesis intermediates are generally reduced in *ole1-20* cells, pointing to a pathway blockage at lanosterol. Why does this occur? One possibility is that lipid saturation also impairs the protein stability or activity of lanosterol demethylase (Erg11), which warrants testing given that the human homologue of Erg11 (LDM) is targeted by the homologue of Doa10 (MARCF6) (Scott et al. 2021 Biochem J).

We have now briefly described the results of Fig. EV3 (now Fig S5) in the results section and discussed the possibility of Erg11 downregulation. In fact, we are currently investigating the issue of Erg11 downregulation upon lipid saturation, hoping to complete the study for another manuscript in the future.

5. Fig. 5B: squalene conversion is restored by the addition of glycerol in Erg1 P469L-expressing *ole1-20* cells; however, there is no corresponding increase in lanosterol levels. Can the authors please comment on this result? As a point of comparison, this experiment would benefit from including cells expressing WT Erg1 (where the ER is intact) to confirm that this indeed averts or blunts lanosterol accumulation.

There is indeed a small increase of lanosterol in Erg1 P469L-expressing *ole1-20* after glycerol addition (Fig. 5B), although it did not correspond to the reduced amount of squalene. It is possible that glycerol also partially alleviates the blockage of lanosterol turnover, such as by preserving Erg11. We have now included this in the discussion section. We have also added in the new Fig 5C the result of sterol analysis from *ole1-20* expressing WT Erg1 or P469L. The data shows that there is only 2-fold

increase of lanosterol in *ole1-20* expressing WT Erg1, supporting our notion that aberrant ER is caused by the large accumulation of lanosterol from Erg1 P469L expression. WT Erg1 is unable to produce the same effect as P469L mutant, likely because it is unstable in *ole1-20*, unlike P469L.

6. Fig. 5D: does manipulating lanosterol levels also restore ER morphology? The authors have only considered cell growth. If so, it would help to support their final model.

This question was also raised by review #1 (point 3). As described above, we now show that Erg1 distribution and ER morphology are indeed restored by Ro 48-8071 in a dose-dependent manner in *ole1-20* overexpressing stable Erg1 (Fig. 5G) and in *ole1-20 doa10Δ* double mutant (Fig. 6E).

Minor comments

1. The statement "both mammalian SQLE and Erg1 are destabilized in response to an elevated level of downstream intermediate lanosterol" is incorrect - degradation of mammalian SQLE is unaffected by lanosterol (Gill et al. 2011 Cell Metab.).

We have corrected the statement.

2. On page 9, it is stated that "Erg1 structure predicted by AlphaFold2 was similar to SQLE (Fig 3A)." However, no comparison between the two proteins is depicted.

We have now included the structure of human SQLE in Fig 3A.

3. Please include methods for the scoring of imaged cells (e.g., Fig. 4B) and tetrad analysis (Fig. 6C).

We have now included the methods.

4. To improve clarity, labels should be added to the icons in Figs. 4A and 6B to explicitly describe the phenotype.

We have now added a few labels in the images to indicate the normal and abnormal patterns. Because the disorganized ER and Erg1 foci are spread throughout the cell, it is hard to label all abnormal structures without messing up and obscuring the pictures. Thus, we only label the representative ones as indicated in the legend.

5. The schematic in Fig. 8 is a useful accompaniment to the discussion, but would benefit from inclusion of Erg1, Doa10 and Ubc7 to better integrate the findings of the entire manuscript. The legend does not outline the possible consequences of protein segregation into different membrane phases - presumably this is the likely cause of the membrane dysfunction?

We have now included Erg1, Doa10 and Ubc7 in Fig 8, and described in the legend the consequence of aberrant ER morphology and mislocalized and/or dysfunctional ER proteins.

6. Please ensure that figure legends contain full information about temperature(s) at which experiments were performed and any medium supplements (e.g., oleic acid, galactose).

We have ensured the inclusion of the information in the legends.

7. Please define the acronyms "TAG" and "NPC" in the main text.

We have defined these terms as they first appear in the main text.

8. Please correct typos in the main text and figures, e.g., "lesser extend" (page 11), "microscopy. Whereas" (page 19), "GFP-P469L" (Fig. 4C), "deleted for" (Fig. 2D legend), "Student's" (Fig. 4C legend).

We have corrected these typos.

Reviewer #3

Minor issues:

P5: When explaining that HMGR becomes misfolded, please cite one of the more recent papers from Randy Hampton, i.e. either Wangeline and Hampton, JBC 2018 or Wangeline and Hampton, JBC 2021.

We have now cited the new paper (Wangeline and Hampton, JBC 2021).

P6: "... lipotoxicity-related diseases (Schaffer, 2016)." Please give examples of the diseases you are referring to.

We have now added diabetes and hepatic steatosis as examples.

P7: "The *ole1Δ* deletion mutant is lethal". The mutant is not lethal but dead. Hence, more accurate ways of phrasing this would be "The *ole1Δ* deletion is lethal" or "The *ole1Δ* deletion mutation is lethal."

The statement is now corrected.

P7: For the benefit of orienting the reader, please begin with a statement why *OLE1* ts alleles were made.

We have now added in the beginning of the results section the rationale of generating *ole1ts* mutants.

P7, Figure 1B: In the figure it says "recovered" instead of recovered.

We have now corrected the typo.

P7, Figure 1D: Please add information to the figure itself that growth is at 34 degrees. This piece of information is in the text and in the figure legend but it would help the reader to also see it in the panel.

We have now added this information in Fig 1D.

P8: The authors mention that SILAC was done to find proteins with differential abundance in WT versus *ole1-20* cells but the data are not provided. Please either include the data or add a comment that or why they are not shown.

We did not show the SILAC data at the first place, because it was not our intention to describe the overall change in the proteome. Nevertheless, we now include a simple scatter plot of the SILAC proteomics in the new Fig S2 to show that Erg1 is reduced in *ole1-20*. We are currently investigating other candidate proteins.

P9: "... showed that Erg1 situated on the surface ...". Please edit to: "... Erg1 is situated on ..."

We have changed the wording accordingly.

P10, Figure 3F: Please use a transmembrane protein as a reference rather than GFP-HDEL. The GFP-HDEL only shows that carbonate and high salt did not disrupt the ER membrane. However, to illustrate that Erg1 is not a regular integral membrane protein, it is necessary to show that the solubility of a true transmembrane protein is unaffected by carbonate treatment.

We have now replaced the control using Sec61 to represent an integral membrane protein.

P11: "... and to a lesser extent at 27 ...", should be: "... and, to a lesser extent, at 27 ..."

We have corrected the wording.

P11: Could the foci of Erg1(P469L) in *ole1-20* cells that co-localize with Erg6 but are not lipid droplets (Figure 4) represent ERAD foci (Albert, ... Engel, PNAS 2020)? Please comment in the discussion if you think that this is a reasonable possibility. This is entirely optional, though, the authors do not have to include this idea if they do not think that it is helpful.

We thank the reviewer for this interesting idea. However, we decide not to consider this possibility at this point, because the ERAD foci were apparently present in *Chlamydomonas reinhardtii* even without ER stress, which might not be relevant to our observation. Also, it is unclear whether ERAD foci also exist in other organisms, including yeast.

P15: "... and for the membrane microdomain of specific functions." Not clear what membrane microdomain is meant here. If you refer to membrane microdomains generally, then edit to: "... for membrane microdomains of specific functions."

We have changed the wording accordingly.

October 24, 2022

RE: Life Science Alliance Manuscript #LSA-2022-01612-TR

Dr. Rey-Huei Chen
Institute of Molecular Biology, Academia Sinica
Institute of Molecular Biology
128 Sec. 2, Academia Rd.
Taipei 11529
Taiwan

Dear Dr. Chen,

Thank you for submitting your revised manuscript entitled "Lipid saturation induces degradation of squalene epoxidase for sterol homeostasis and cell survival". We would be happy to publish your paper in Life Science Alliance pending final revisions necessary to meet our formatting guidelines.

-please include the supp. Material and Methods section into the main Materials and Methods section

Figure Check:

- Figure 1C need scale bars
- Figure 5 A, E, F, Figure 6D need scale bars

A. FINAL FILES:

B. MANUSCRIPT ORGANIZATION AND FORMATTING:

Sincerely,

Reviewer #1 (Comments to the Authors (Required)):

In this revised manuscript Huang and Chen explore the effects of phospholipid saturation on stability of a model membrane protein in yeast, Erg1. To control saturation, they generated temperature-sensitive mutants in the desaturase Ole1, and they obtained a mutant that accumulates saturated fatty acids at high temperature that is toxic to cell survival. Authors found that Erg1, a protein in the ergosterol biosynthetic pathway, is unstable stable in this ole1ts mutant at high temperature. Supplementation with oleic acid (OA) yields stable Erg1. They found that Doa1 and Ubc7, proteins of the ERAD complex, is involved in the degradation of Erg1 upon lipid saturation. The authors then identified Erg1 mutants resistant to degradation upon lipid saturation. They determined cell localization of Erg1 (normally in the ER) and found that it forms foci upon lipid saturation, and that ER morphology was aberrant. The toxic effects of saturation on cells were attributable to these changes in ER morphology, which, in turn, was caused by accumulation of lanosterol. Finally, they demonstrated that addition of glycerol resulted in normal ER morphology and prevention of protein misfolding, presumably due to a chaperone-like effect of this carbon source.

These data are significant in that they establish a new model for tightly temporally regulating phospholipid saturation in yeast, and they demonstrate that folding and stability of membrane proteins, as well as organellar morphology, are exquisitely dependent on this membrane parameter. The paper is easy to read, and the data are appropriately interpreted.

I offered some criticisms of the original manuscript. All were sufficiently dealt with in this revision. I have no further critical comments.

Reviewer #2 (Comments to the Authors (Required)):

Congratulations on a very nice piece of work.

Reviewer #3 (Comments to the Authors (Required)):

The minor issues I raised in my original comments have been addressed adequately and I support publication of the manuscript as is.

October 25, 2022

RE: Life Science Alliance Manuscript #LSA-2022-01612-TRR

Dr. Rey-Huei Chen
Institute of Molecular Biology, Academia Sinica
Institute of Molecular Biology
128 Sec. 2, Academia Rd.
Taipei 11529
Taiwan

Dear Dr. Chen,

Thank you for submitting your Research Article entitled "Lipid saturation induces degradation of squalene epoxidase for sterol homeostasis and cell survival". It is a pleasure to let you know that your manuscript is now accepted for publication in Life Science Alliance. Congratulations on this interesting work.

DISTRIBUTION OF MATERIALS:

Again, congratulations on a very nice paper. I hope you found the review process to be constructive and are pleased with how the manuscript was handled editorially. We look forward to future exciting submissions from your lab.

Sincerely,
